# Identification and characterization of a direct activator of a gene transfer agent

Paul C.M. Fogg [1]

Gene transfer agents (GTAs) are thought to be ancient bacteriophages that have been co-opted into serving their host and can now transfer any gene between bacteria. Production of GTAs is controlled by several global regulators through unclear mechanisms. In *Rhodobacter capsulatus*, gene *rcc01865* encodes a putative regulatory protein that is essential for GTA production. Here, I show that *rcc01865* (hereafter *gafA*) encodes a transcriptional regulator that binds to the GTA promoter to initiate production of structural and DNA packaging components. Expression of *gafA* is in turn controlled by the pleiotropic regulator protein CtrA and the quorum-sensing regulator GtaR. GafA and CtrA work together to promote GTA maturation and eventual release through cell lysis. Identification of GafA as a direct GTA regulator allows the first integrated regulatory model to be proposed and paves the way for discovery of GTAs in other species that possess *gafA* homologues.

[1] Biology Department, University of York, Wentworth Way, York YO10 5DD, UK. Correspondence and requests for materials should be addressed to P.C.M.F. (email: paul.fogg@york.ac.uk)

Rapid bacterial evolution is a fundamental process that allows bacteria to adapt to changes in their environment and to explore new environmental niches. The primary mechanisms for the rapid spread of genes are known collectively as Horizontal Gene Transfer (HGT). In contrast to hereditary transfer, HGT allows genes to be passed directly between individual bacteria at a much faster rate[1,2]. The genes being transferred may improve fitness or resilience but can also lead to antimicrobial resistance (AMR) or increased virulence.

Traditionally, bacterial HGT consists of three broad mechanisms of genetic exchange – conjugation, transformation and transduction. Transduction by bacteriophages is generally accepted to be the most influential mechanism for the exchange of genes between bacteria, in particular, the generalized transducing (GT) phages and the recently described lateral transducing (LT) phages play a crucial role[3]. During phage replication, host bacterial DNA is packaged into a significant proportion of phage particles instead of the phage genome; the host DNA can be randomly selected (GT phages) or it can be from a large hypermobile region (LT phages). The packaged host DNA is then protected by the phage capsid and delivered to a new host cell, where it can be integrated into the target genome by homologous recombination.

Gene transfer agents (GTAs) are an unusual method of HGT, which appears to be a hybrid of bacteriophage transduction and natural transformation[4]. First discovered in the 1970s, GTAs are small virus-like particles that transfer random fragments of the entire genome of their bacterial host between cells[5]. Unlike the transducing phages, whose primary aim is still self-preservation, GTAs have no preference for the spread of their own genes and their survival is entirely dependent upon their hosts' wellbeing[6,7]. It is the complete lack of DNA selectivity that makes GTAs particularly intriguing and raises important questions about their impact on HGT, bacterial evolution and the selective pressures that allow them to persist[8].

A rough estimate of the number of viruses in the oceans alone is $4 \times 10^{30}$ ref. [9]. Metagenomic analyses of the marine virome typically reveal that >60% of the sequences are unrelated to any known viruses, and there has been speculation that GTAs are a significant contributor to this cloud genome[10,11]. A seminal study of antibiotic gene transfer by GTAs in in situ marine microcosms, observed frequencies that were orders of magnitude greater than any known mechanism[12]. In the model host, *Rhodobacter capsulatus*, RcGTAs are under the control of a number of conserved global regulatory systems such as the cell cycle regulator CtrA[13–15], the quorum-sensing regulator GtaR[16,17] and various phosphorelay components such as DivL and CckA[15,18], however, all of these regulators affect RcGTA production indirectly and thus the mechanism of activation is unclear.

In this study, I identify and characterize a transcription factor (Rcc01865, renamed GafA here) that binds directly to the RcGTA promoter. The *gafA* promoter is in turn bound by both the pleiotropic regulators CtrA and GtaR near the transcription start site. CtrA and GafA are both required for optimal RcGTA expression, packaging of DNA and release of infective particles. The data presented here indicates that GafA is the missing link that connects RcGTA production with host regulatory systems and allows construction of the most comprehensive model of RcGTA regulation to date.

## Results and Discussion
### All RcGTA genes are upregulated in an RcGTA hyperproducer.
RcGTAs are usually produced from a small sub-population, making in-depth analysis of RcGTA producers problematic[6,19].

Here we compared the transcriptome of an RcGTA hyperproducer, *R. capsulatus* DE442, to the wild-type by RNAseq[19]. 152 upregulated and 37 down regulated genes were identified (Supplementary Tables 1 & 2). The top 29 upregulated genes had a beta value (*b*) of 4.0 or greater (Supplementary Table 3), equivalent to a 16-fold increase in transcript abundance, and contained all of the genes from the core RcGTA structural gene cluster[14], head spikes[20], tail fibre[21], lysis genes[18] and a putative RcGTA maturation protein[22]. One further gene, *rcc01865*, was previously shown to be essential for RcGTA production but its precise role is unknown[22]. *Rcc01865* encodes a protein with a predicted helix-turn-helix (HTH) DNA binding motif in the N-terminal domain that structurally resembles the DNA binding domain (DBD) of the genome replication initiator protein DnaA (e.g. *Mycobacterium tuberculosis* DnaA-DBD, 3PVV; Supplementary Figure 1), which led to the assumption that it is a regulator protein[22]. The C-terminus contains a region that has similarity to various sigma factors, including a high HHPRED probability match to *Rhodobacter sphaeroides* RpoE (Supplementary Figure 1). Given that *rcc01865* is essential for RcGTA production[22] and encodes the only putative transcription factor in the top 29 upregulated genes in the RNAseq data (Supplementary Table 3), it is a strong candidate to be a specific initiator of RcGTA production. *Rcc001865* will hereafter be referred to as GTA Activation Factor A (*gafA*).

### GafA activates production of RcGTA particles.
Deletion of *gafA* completely prevents RcGTA gene transfer[22], even in the hyperproducer strain *R. capsulatus* DE442 (Fig. 1a) where RcGTA gene expression, gene transfer frequencies and the proportion of the producing RcGTAs are normally substantially increased[6,19]. Furthermore, in DE442, packaged GTA DNA can be seen as a distinct 4 kb band in a total DNA purification. Deletion of *gafA* prevents any detectable GTA DNA in this assay (Fig. 1b), indicating that RcGTA production is fundamentally undermined at or before the DNA packaging stage. Overexpression of *gafA* in wild-type *R. capsulatus* SB1003 increased antibiotic gene transfer frequencies 57-fold (SD = 7, $n = 8$), compared to 94-fold for the stable hyperproducer phenotype (SD = 19, $n = 8$) (Fig. 1a)[19]. In addition, total DNA from the *gafA* overexpressor contained large quantities of 4 kb GTA DNA after 6 h (Fig. 1b). After 24 h, the cells partially dampened RcGTA production, although the levels observed were still far greater than WT (Fig. 1b). Dampening of RcGTA production is not unexpected as uniform expression in all cells is likely to be highly deleterious[6,18,19,23].

### CtrA overexpression does not lead to RcGTA overproduction.
Previous work showed that the global regulator protein CtrA is also essential for RcGTA production[14], however, the mechanism has never been discovered. Similar to *gafA*, deletion of *ctrA* prevents any detectable RcGTA gene transfer or production of the RcGTA capsid protein[14]. Activity of CtrA is modulated by phosphorylation of an aspartic acid residue (D51), and its phosphorylation state is important for RcGTA production[15,24]. The RNAseq data showed that CtrA is upregulated (2.5-fold) in DE442 (Supplementary Tables 1 & 3) along with known CtrA regulon genes for chemotaxis and motility (Supplementary Table 1). If *gafA* is a simple constituent of the CtrA regulon then increasing the abundance CtrA should lead to RcGTA overproduction. Overexpression of WT *ctrA* or phosphomimetic *ctrA*[D51E] led to a slight reduction in RcGTA gene transfer, whereas non-phosphorylatable *ctrA*[D51A] increased gene transfer 2-fold (Fig. 2a)[25]. No GTA DNA bands were detected in total DNA for any of the *ctrA* overexpressor strains (Fig. 2b), which was consistent with no effect or a modest increase in RcGTA

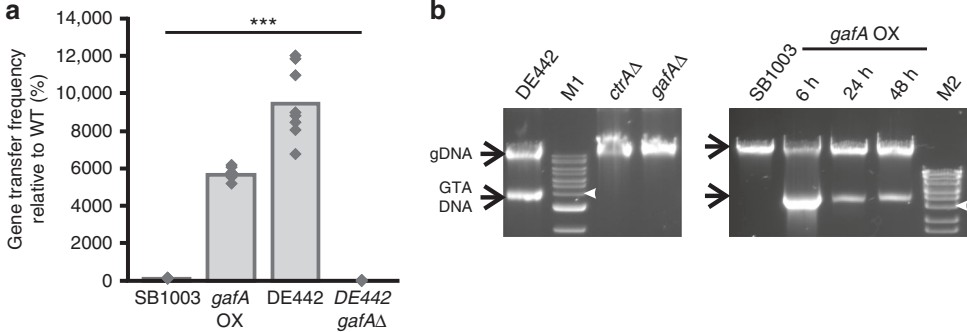

**Fig. 1** Confirmation of the RcGTA Activator, GafA. **a** GTA gene transfer assays for *R. capsulatus* SB1003 (WT), SB1003 *gafA* overxpressor (*gafA* OX), RcGTA hyperproducer strain *R. capsulatus* DE442 (DE442) and DE442 with *gafA* deleted (DE442 *gafA*Δ). Individual replicates are shown as diamonds. All conditions were significantly different; One Way ANOVA significance is indicated above the bars ($n = 8$, *** = $p < 0.001$). **b** Agarose gels of total DNA isolated from the annotated *R. capsulatus* strains - RcGTA hyperproducer strain *R. capsulatus* DE442, *ctrA* (*ctrA*Δ) and *gafA* (*gafA*Δ) knockouts in DE442, wild-type *R. capsulatus* SB1003 compared to *gafA* overexpressor (OX) derivatives of SB1003. Time post induction of *gafA* is noted in hours, GTA and genomic DNA (gDNA) are indicated by labelled arrows. NEB 1 kb Extend DNA Ladder (M1) or Bioline HyperLadder 1 kb DNA ladder were used (M2); the 4 kb band is annotated with a white arrow head. Source data are provided as a Source Data file

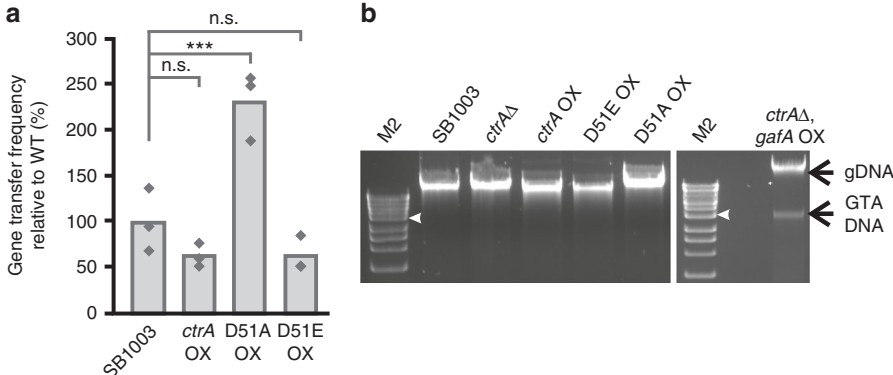

**Fig. 2** The role of CtrA in RcGTA production. **a** GTA gene transfer assays for *R. capsulatus* SB1003, *ctrA* overexpressor (*ctrA* OX), non-phosphorylatable *ctrA* overexpressor (D51A) and phosphomimetic *ctrA* overexpressor (D51E OX). Individual replicates are shown as diamonds ($n = 3$), One Way ANOVA significance versus the control (SB1003) is indicated above the chart (n.s. not significant i.e. $p > 0.05$, ***$p < 0.001$). **b**, Agarose gels of total DNA isolated from *R. capsulatus* SB1003 and the annotated derivatives – wild-type *R. capsulatus* SB1003, *ctrA* knockout (*ctrA*Δ), *ctrA* overexpressor (*ctrA* OX), phosphomimetic *ctrA* overexpressor (D51E OX), non-phosphorylatable *ctrA* overexpressor (D51A OX) and a *gafA* overexpressor in a *ctrA* knockout background (*ctrA*Δ, *gafA* OX). GTA and genomic DNA (gDNA) are indicated by labelled arrows. Bioline HyperLadder 1 kb DNA ladder was used (M2); the 4 kb band is annotated with a white arrow head. Source data are provided as a Source Data file

production. Similar to the *gafA* deletion, *ctrA* knockouts were not able to produce any detectable RcGTAs in WT[13,14] or hyperproducer strains (Fig. 2).

**CtrA controls GafA activity, RcGTA maturation and lysis**. Overexpression of *gafA* in cells lacking *ctrA* still led to substantial intracellular GTA DNA accumulation (Fig. 2b), albeit at a lower level than in *ctrA* replete cells (Fig. 1b), indicating that the essential role of CtrA in expression of the GTA structural gene cluster is upstream of GafA. Overexpression of *gafA*, however, did not rescue RcGTA gene transfer ability in the *ctrA* knockout, DNaseI insensitive DNA was not detectable in the culture supernatant and manual lysis of the cells did not release any detectable infective RcGTA particles. Taken together, these data show that GafA activates synthesis of the RcGTA structural genes and packaging of host DNA, whilst, CtrA is required for maturation and release of infective RcGTA particles.

To further investigate the relationship between CtrA, GafA and RcGTA production, transcription of various GTA-related genes was measured. As expected from the phenotypic profiles, deletion

of *ctrA* or *gafA* in DE442 eliminated the hyperproducer expression profile. Expression of the RcGTA terminase, capsid and endolysin genes all reduced to basal levels (Fig. 3a). Deletion of *ctrA* also reduced *gafA* expression but deletion of *gafA* did not affect *ctrA* expression, which was consistent with the hypothesis that *gafA* is part of the CtrA regulon.

Overexpression of *ctrA* did not lead to a substantial increase in transcription of the RcGTA structural genes, lysis cassette or *gafA* (Fig. 3b), but did increase the abundance of native *ctrA* transcripts indicating positive autoregulation (Fig. 3c). Overexpression of *gafA* in WT cells led to a large increase in RcGTA gene expression (Fig. 3d). After 6 hours, *gafA* was overexpressed 34-fold leading to a large increase in terminase (78-fold), capsid (6-fold) and endolysin (6-fold) transcripts, supporting the hypothesis that GafA is an activator of core RcGTA gene expression and is also involved in the endgame of RcGTA release. In the *ctrA* knockout, overexpression of *gafA* was even greater (198-fold) with an associated increase in terminase (126-fold) and capsid (22-fold) transcription but endolysin upregulation was diminished (Fig. 3d). Lack of lysis in the absence of *ctrA* is a likely explanation for increased transcript abundance for *gafA* and the

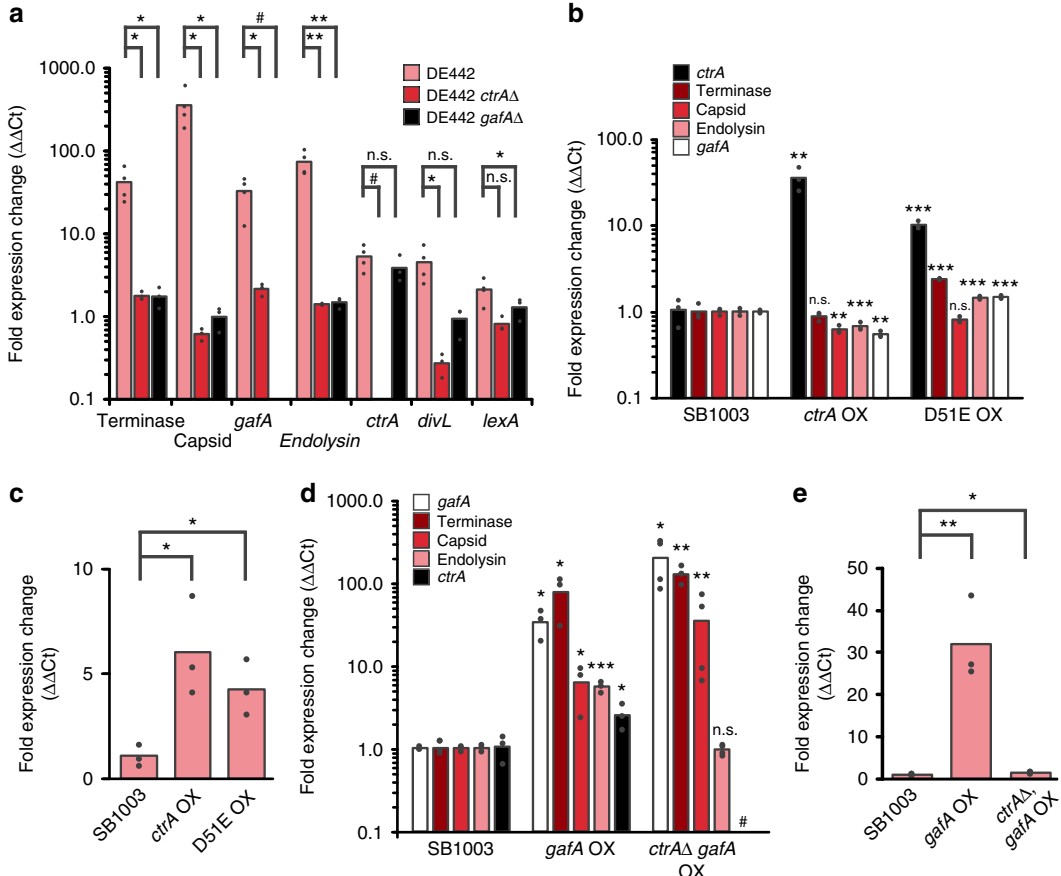

**Fig. 3** Relative Transcription of RcGTA-Related Genes. The *R. capsulatus* strains and gene targets assessed are annotated on each graph. OX indicates a gene overexpressor and Δ is a gene knockout. All Y-axis fold expression changes are normalized using *uvrD* as an endogenous reference gene (ΔCt) and relative to the wild-type SB1003 strain (ΔΔCt). Dot plots of individual replicates are overlaid onto each bar (biological replicates, $n \geq 3$ for all samples). Statistical significance was determined using a two-tail t-test (*$p < 0.05$, **$p < 0.01$, ***$p < 0.001$, #transcript not detected in knockout lines, n.s. not significant i.e., $p > 0.05$). Total transcripts were measured in (**a**, **b**, **d**) and transcripts originating from the native promoter only in (**c**, **e**). Source data are provided as a Source Data file

RcGTA genes. The requirement of CtrA for endolysin production is presumably to allow temporal control of the different stages of RcGTA production, e.g. lysis must not occur before RcGTA particles are fully mature and infective. Transcription of *gafA* from the native promoter also increased 31-fold in response to ectopic *gafA* expression (Fig. 3e). Strong positive *gafA* autoregulation could represent a hair trigger that, once initiated, locks the cell into a lytic fate. In contrast, only a 1.5-fold increase in native *gafA* transcripts was detected in the absence of *ctrA* (Fig. 3e). These data clearly indicate that GafA induces expression of the core RcGTA genes independent of CtrA, however, positive autoregulation of its own transcription is CtrA dependent, providing further evidence that CtrA is required for activation of GafA. Meanwhile, given that deletion of either *ctrA* or *gafA* in DE442 downregulates endolysin expression and GafA only induces endolysin expression in *ctrA* replete cells, both CtrA and GafA must act in concert to promote lytic release of RcGTAs.

**LexA and DivL are upregulated in RcGTA overproducers.** In other species such as *Caulobacter crescentus*, *ctrA* is an essential cell cycle regulator[25,26] and in *Rhodobacter*, although not essential, it must control the timing of distinct phases of RcGTA production. Recent work identified a phosphorelay (ChpT/CckA/DivL) that modulates CtrA phosphorylation[15,18] and dysregulation of the PAS/PAC domain protein DivL led to increased

RcGTA production[15]. *DivL* transcript abundance was 4 to 7-fold upregulated in DE442 (Fig. 3a and Supplementary Table 3) but unaffected by *gafA* overexpression and mildly increased by *ctrA* overexpression (Supplementary Figure 2A). *DivL* was, however, significantly down regulated in *ctrA* knockouts (Supplementary Figure 2A). The SOS repressor, *lexA*, is also required for efficient RcGTA production by regulating the production of CckA[27]. *GafA* and *ctrA* overexpression both led to a marginal increase (1.5 to 2-fold) in *lexA* transcription and, in DE442, *lexA* transcripts were 2 to 8-fold higher than WT (Fig. 3a, Supplementary Figure 2B and Supplementary Table 3). It is likely that a moderate increase in LexA represses CckA, which in turn shifts the CtrA equilibrium toward the unphosphorylated state and thus boosts RcGTA production[27].

**CtrA binds near the *gafA* transcription start site.** Clearly, CtrA and GafA work together to control RcGTA production. There is an obvious CtrA binding site in its own promoter (GTAAC-$N_6$-TTAAC, Fig. 4a) and the GafA promoter contains an almost identical sequence (TTAAC-$N_6$-GTAAC, Fig. 4a)[13,28]. Alignment of the *R. capsulatus gafA* promoter with *gafA* promoters from 14 different species (Supplementary Figure 3), revealed remarkable conservation of the CtrA binding site and its distance to the start codon (usually 65–71 bases) despite otherwise divergent flanking sequences. In an electrophoretic motility shift assay (EMSA),

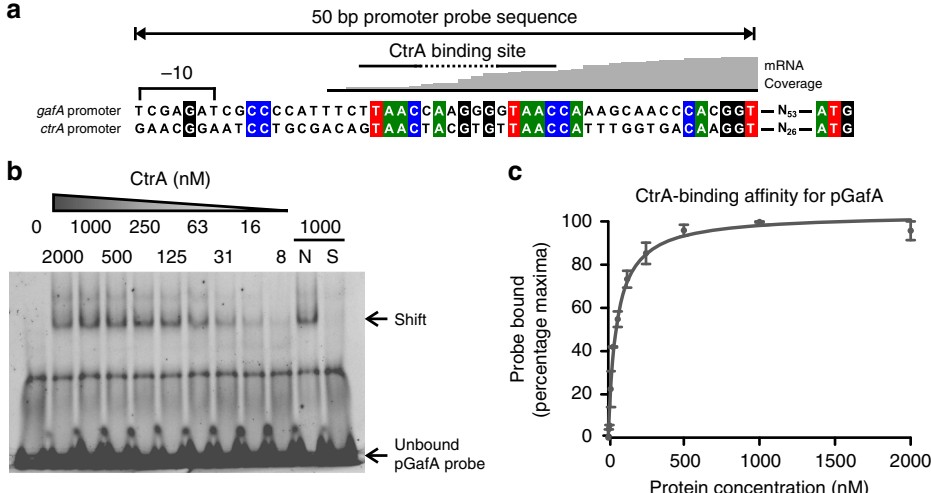

**Fig. 4** CtrA binding to the *gafA* promoter. **a** Alignment of the DNA probe sequences containing CtrA-binding sites that were used for EMSAs (double headed arrow). CtrA binding sites are annotated with half sites represented by solid lines and the spacer sequence as a dashed line. The predicted Shine Delgarno −10 site is annotated. mRNA transcript coverage for the *gafA* promoter, obtained from RNAseq data, is shown as a histogram above the alignment. **b** EMSA band shift of Cy5-labelled *gafA* promoter DNA incubated with the protein concentrations specified. The lane labelled N contained 500-fold excess of an unlabelled non-specific competitor and S contained 500-fold excess of an unlabelled specific competitor. **c** Quantification of two independent band shifts of CtrA vs. the *gafA* promoter. Error bars are standard deviation, $n = 2$. Source data are provided as a Source Data file

purified CtrA had no detectable binding affinity for its own promoter (≤8000 nM Protein, Supplementary Figure 4A), however, CtrA$^{D51E}$ was able to bind to the promoter at low affinity (Supplementary Figure 4B). In contrast, CtrA bound to the *gafA* promoter with much greater affinity than the *ctrA* promoter (Kd 54.91 nM, SD 6.12, Fig. 4b, c), in agreement with the observations that CtrA is essential for *gafA* transcription. Furthermore, the hypothesis that CtrA regulates *gafA* transcription was strengthened by mapping raw RNAseq transcript reads onto the *gafA* promoter sequence, which revealed that the transcription start site is likely to be ~87 bp upstream of the start codon and coincides with the CtrA binding site (Fig. 4a). To test whether CtrA binding to the *gafA* promoter is required for RcGTA production, SB1003 *gafAΔ* was complemented *in trans* with plasmids containing either *gafA* expressed from its unaltered native promoter (pCMF180) or with either of the two CtrA binding half-sites mutated by site directed mutagenesis (pCMF214 and pCMF215) (Supplementary Figure 5). Complementation with the wild-type promoter construct increased gene transfer frequency to 337% of WT (SD = 2%, $n = 3$, ANOVA *p* value =<0.001), presumably due to increased copy number of the plasmid borne *gafA*, whereas both mutated promoter constructs were significantly impaired for gene transfer (10–22% of WT, $n = 3$, ANOVA *p* value = < 0.001).

**The quorum-sensing regulator GtaR binds the *gafA* promoter.** CtrA is evidently important for GafA production, however, it is unlikely to be the only regulator acting on *gafA*. CtrA is expressed throughout all growth stages, whereas RcGTA are only produced in stationary phase[5,29], and its expression is homogenous in wild-type cells[30], whereas RcGTA are only produced by <1% of the population[6,19]. Moreover, overexpression of *ctrA* does not lead to a substantial increase in *gafA* transcription or RcGTA production (Figs. 2 and 3). The GtaI/R quorum-sensing system is also essential for RcGTA production[16,17,31]. Regulation by quorum-sensing would certainly allow *gafA* and RcGTA expression to be limited to stationary phase and heterogeneity of the response to homoserine lactone inducer signal could also be responsible for RcGTA phase variation[32–34]. Band shifts were carried out using

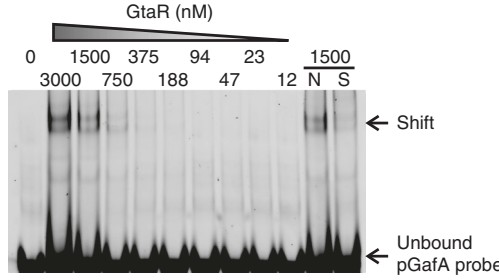

**Fig. 5** Binding of the GtaR quorum-sensing protein to the *gafA* promoter. EMSA band shift of Cy5-labelled *gafA* promoter DNA (see Fig. 4a) incubated with the protein concentrations specified. The lane labelled N contained 500-fold excess of an unlabelled non-specific competitor and S contained 500-fold excess of an unlabelled specific competitor. Source data are provided as a Source Data file

the same *gafA* promoter region that contains the CtrA binding site (Fig. 4a) and purified GtaR. GtaR binding was detected at concentrations of 375 nM or above (Fig. 5). The only known binding site for GtaR is within its own promoter[16] and no analogous sequence was detected in the 50 bp promoter fragment used here, which is not unexpected. Binding sites for quorum-sensing proteins are thought to be highly degenerate and thus difficult to predict; indeed Leung et al. (2013) reported that the best matches to the model GtaR binding site in *R. capsulatus* were not bound in vitro[16]. It is notable that GtaR binds to its own promoter at a location spanning the predicted -10 Shine Delgarno element and the transcription start site[16], and the *gafA* promoter region bound by GtaR here contains the same promoter features (Fig. 4a).

**GafA, but not CtrA, binds to the RcGTA promoter.** The data presented so far suggest that GafA acts as a direct regulator of RcGTA expression and it is likely to bind to the promoter region of the structural gene cluster, hereafter referred to as the RcGTA

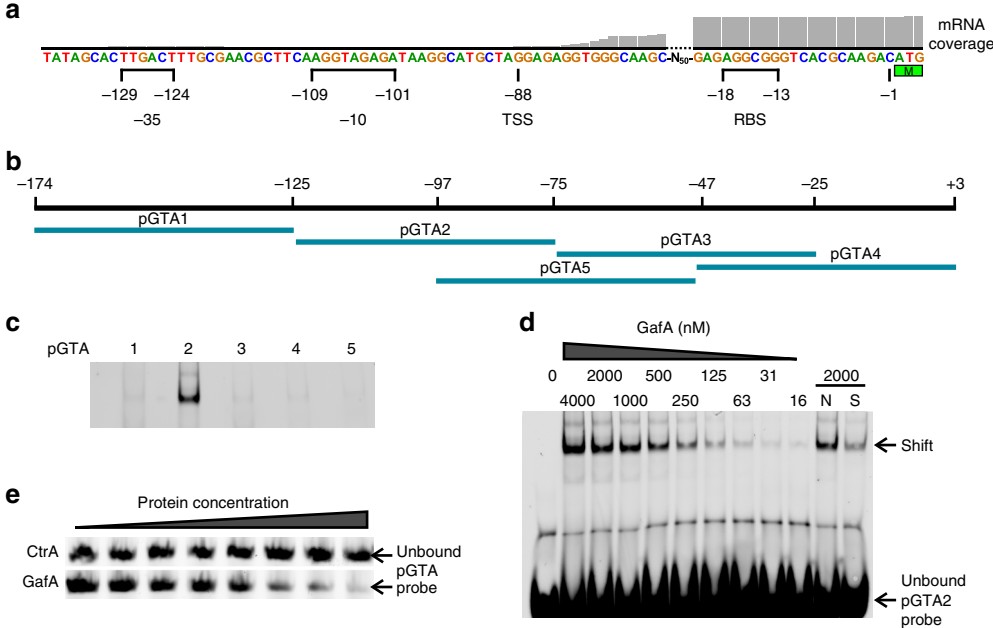

**Fig. 6** GafA binding to the RcGTA cluster promoter. **a** Map of the RcGTA structural gene cluster promoter indicating the predicted locations of the Shine Delgarno −10 and −35 sites, the ribosome-binding site (RBS), RcGTA *g1* start codon and transcription start site (TSS). mRNA transcript coverage, obtained from RNAseq data, is shown as a histogram. **b** Map of the overlapping 50 bp regions of the RcGTA promoter used as EMSA probes (pGTA1-5). **c** EMSA band shifts of Cy5-labelled pGTA1-5 vs. 2 μM GafA protein. **d** EMSA band shift of titrated GafA protein at the concentrations indicated versus Cy5-pGTA2. The lane labelled N contained 500-fold excess of an unlabelled non-specific competitor and S contained 500-fold excess of an unlabelled specific competitor. **e** Unshifted Cy5 labelled, 633 bp RcGTA promoter DNA after incubation with up to 4 μM of either CtrA or GafA. Source data are provided as a Source Data file

promoter. The RcGTA promoter is not well characterized and no transcription factors have been identified that bind in this region. An EMSA was carried out with five overlapping 50 bp probes that were designed to cover the 174 bp region immediately upstream of RcGTA *g1* (Fig. 6a, b). GafA binding was only detected with one of the five probes (pGTA2, Fig. 6c) spanning the region 76–125 bp upstream of the RcGTA *g1* start codon (Fig. 6a). Titration of the GafA protein revealed detectable binding to pGTA2 with as low as 16 nM protein (Fig. 6d). Accurate estimation of the Kd was not possible because there were insufficient data points at full saturation, however, it is likely to be in the high nanomolar range. The pGTA2 promoter region contains the predicted -10 element and the transcription start site, which was confirmed by analysis of the raw RNAseq mRNA coverage (Fig. 6a). Binding of GafA to the region containing the −10 and TSS, together with phenotypic and qPCR data described above, strongly supports the hypothesis that GafA is a direct regulator of RcGTA at the transcriptional level, possibly as an alternative sigma factor. Mercer et al. (2014) reported a putative partner switching signalling pathway, comprising RbaV, RbaW and RbaY, that when disrupted had a moderate but significant effect on RcGTA production (<3-fold)[24]. RbaW was predicted to be an anti-sigma factor and extensive attempts were made to identify the cognate sigma factor, including deletion of all known sigma factors except RpoN and RpoD, none of which were found to interact with RbaW or affect expression of RcGTA. GafA had not been linked to RcGTA at that time and thus was not considered, but it is possible that GafA is the target of RbaW.

Meanwhile, no CtrA binding was detected to the full length RcGTA promoter (Fig. 6e), confirming that CtrA regulation is indirect. The data presented are the first evidence of a transcription factor activating a GTA promoter and for the first time a direct link has been established with core host regulatory

pathways via CtrA and GtaR. Furthermore, GafA binds to its own promoter region (Supplementary Figure 6A) to positively auto-regulate its own expression (Fig. 3e) and to the lysis cassette promoter (Supplementary Figure 6B) to induce endolysin expression (Fig. 3d), indicating that GafA plays a critical role in both RcGTA production and subsequent release.

**GafA is a core component of an RcGTA-regulation model.** The results presented here allow a model of RcGTA regulation to be constructed (Fig. 7). *Rhodobacter* RcGTA production begins in stationary growth phase, controlled by the quorum-sensing protein[16]. Once RcGTA production begins, unphosphorylated CtrA activates *gafA* expression; GafA then enhances its own expression, activates expression of the core GTA structural cluster and packaging of DNA into capsids. GTAs are normally produced in a small proportion of any given population[6,19,35], however, in wild-type cells CtrA expression is more or less homogenous[30] and simple overexpression of *ctrA* does not lead to high level expression of *gafA* (Fig. 2), which suggests that there are other unknown factors in play. There is no evidence that epigenetic factors, such methylation or DNA inversions, influence RcGTA production but heterogeneity in the quorum-sensing response is a possible explanation for RcGTA phase variation. Relative fitness has been implicated as a factor that induces Bartonella GTA (BaGTA)[35], i.e. the fittest subpopulation spontaneously produce BaGTAs presumably to spread the most beneficial genes, but contradictory data has been reported for RcGTA suggesting that it is starvation that leads to production[18,27,36]. Subsequent to induction of the RcGTA structural genes, CtrA is phosphorylated by the DivL/CckA/ChpT phosphorelay[15]. CtrA-P activates expression of maturation and secondary structural proteins required for infectivity[15]. Finally, GafA binds to the endolysin

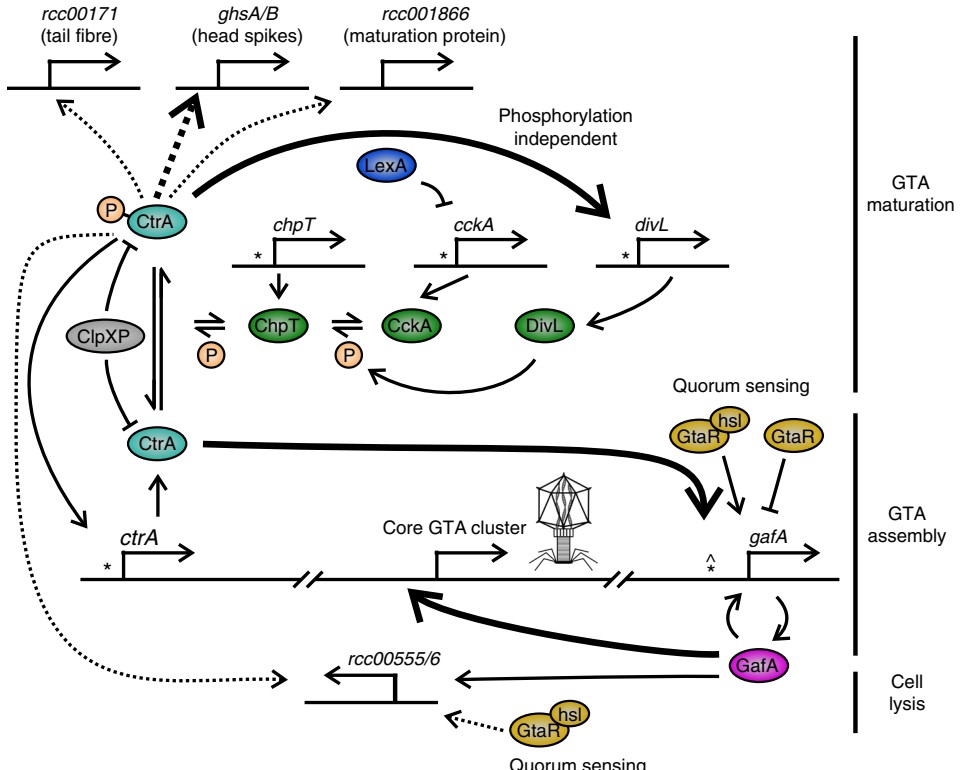

**Fig. 7** Model of RcGTA regulation. The interactions depicted are inferred from the data in this study, raw microarray data[16] and published results[18,19,25,30]. Bent, perpendicular arrows represent promoters and are annotated with the proceeding gene name. CtrA (*) or GtaR (^) binding sites are labelled where known. Proteins are depicted as coloured ellipses with phosphate groups (P) in orange circles. Solid arrows indicate direct regulation, dashed arrows indicate indirect or unknown route of regulation and emboldened arrows indicate that the regulator is essential for target expression

promoter and induces CtrA-dependent cell lysis and RcGTA release.

Hynes et al.[22] reported that GafA homologues are present throughout the Rhodobacterales, including in each of the confirmed GTA producers, and local synteny of GafA is broadly conserved i.e. it is usually flanked by lipoyl synthase (*lipA*) and GMP synthase (*gua1*) genes[22]. Overexpression of *gafA* homologues from two known GTA producers (*Ruegeria mobilis* & *Roseovarius nubinhibens*, Supplementary Figure 7) also led to increased GTA production (Supplementary Figure 8), demonstrating that activation of GTAs by GafA is not unique to *R. capsulatus*. Although GafA is present in various different species, its rate of evolution was reported to be faster than most components of the RcGTA genome, albeit only marginally so[22]. In general, all RcGTA genes tend to be evolving faster than core host genes and slower than comparable phage genes[22]. Beyond the Rhodobacterales, *gafA* homologues can be found in the Rhizobiales[37], a bacterial order that includes plant and animal pathogens such as *Agrobacterium tumafaciens* and *Brucella abortus*. Rhizobiales *gafA* genes are usually share less than 25% homology with their Rhodobacterales counterparts[37] or are split into two separate ORFs, for example in *A. tumafaciens* (NZ_ASXY01000077) each ORF product is homologous to the either the N-terminal DnaA DBD-like domain or C-terminal sigma factor-like domains.

GTAs are thought to be derived from ancient bacteriophage that have been hijacked by their host[22], although the lack of significant matches to GTA genes in α-proteobacterial CRISPR spacer regions suggest that the hypothetical progenitor phage is extinct[37]. Several marine Roseophages, such as RDJLΦ1, contain several GTA-like structural genes as well as both GafA and its

neighbour, rcc01866[22,38], but they are separated by a single intervening gene with clear homology to CtrA[7]. The phage version of CtrA lacks the N-terminus, which contains the response regulator domain, but retains the transcriptional activator domain. The presence of homologues of essential RcGTA regulator and structural genes in a phage suggests that the relationship between these regulators and GTA production is ancient.

GTAs have the potential to drive bacterial evolution and genome plasticity, including the spread of virulence and AMR genes. Here, GafA is identified as the first direct activator of GTA expression to be reported for any species. The data allow the construction of a comprehensive model of RcGTA regulation that brings together the roles of the pleiotropic regulator CtrA, quorum sensing, the SOS response and a conserved phosphorelay chain. Furthermore, many aspects of GTA biology make them intractable for high throughput studies, but identification of direct activators of GTAs in widespread species could open up a new frontier in GTA research.

## Methods

**Bacterial strains**. Two wild-type *Rhodobacter* strains were used – rifampicin resistant SB1003 (ATCC BAA-309) and rifampicin sensitive B10[39]. The RcGTA overproducer strain DE442 is of uncertain provenance but has been used in a number of RcGTA publications[19,40]. The *E. coli* S17-1 strain, which contains chromosomally integrated *tra* genes, was used as a donor for all conjugations. NEB 10-beta Competent *E. coli* (New England Biolabs, NEB) were used for standard cloning and plasmid maintenance; T7 Express Competent *E. coli* (NEB) were used for overexpression of proteins for purification. *Ruegeria mobilis* (DSM 23403), *Roseovarius nubinhibens* (DSM 15170) and *Ruegeria pomeroyi* (DSM 15171) are reported GTA producers that were all obtained from DSMZ. All bacterial strains or genetic constructs are securely stored locally and are available on request.

**Cloning**. All oligonucleotides were obtained from IDT (Supplementary Table 4) and designed with an optimal annealing temperature of 60 °C when used with Q5 DNA Polymerase (NEB). All cloning reactions were carried out with either the In-Fusion Cloning Kit (CloneTech) or NEBuilder (NEB) to produce the constructs listed in Supplementary Table 5. In summary, destination plasmids were linearized using a single restriction enzyme (pCM66T (BamHI), pEHisTEV (NcoI) and pSRKBB (NdeI)) or by PCR (pETFPP_2 using primers CleF and CleR). Inserts were amplified using primers with 15 bp 5′ overhangs that have complementary sequence to the DNA with which it is to be recombined.

**Transformation**. Plasmids were introduced into all species except *Rhodobacter* by transformation. *E. coli* was transformed by standard heat shock transformation[41]. For *Ruegeria* and *Roseovarius*, 200 ml cultures were washed three times in ice cold 10% glycerol (100 ml then 50 ml then 5 ml). 100 μl aliquots were mixed with 100 ng plasmid DNA and incubated on ice for 30 min. Electroporation was carried out in 2 mm electroporation cuvettes (Scientific Laboratory Supplies) at 2.5 kV, 25 μF and 100 Ω. 1 ml of marine broth was added and cells incubated at 30 °C for 4 h, then plated onto MB agar + 50 μg ml⁻¹ kanamycin.

**Conjugation**. One millilitre aliquots of overnight cultures of the *E. coli* S17-1 donor and *Rhodobacter* recipient strains were centrifuged at $5000 \times g$ for 1 min, washed with 1 ml SM buffer, centrifuged again and resuspended in 100 μl SM buffer. Ten microlitres of concentrated donor and recipient cells were mixed and spotted onto YPS agar or spotted individually as negative controls. Plates were incubated o/n at 30 °C. Spots were scraped, suspended in 100 μl YPS broth and plated on YPS + 100 μg ml⁻¹ rifampicin (counter-selection against *E. coli*) + 10 μg ml⁻¹ kanamycin (plasmid selection). Plates were incubated o/n at 30 °C then restreaked onto fresh agar to obtain single colonies.

**Nucleic acid purification**. One millilitre samples of relevant bacterial cultures were taken for each nucleic acid purification replicate. Generally, sampling occurred during stationary phase but for overexpression experiments samples were taken 6 h and 24 h after transition to anaerobic growth. Total DNA was purified according to the Purification of Nucleic Acids by Extraction with Phenol:Chloroform protocol[41]. In brief, cell pellets were resuspended in 567 μl TE buffer then 30 μl of 10% SDS and 3 μl of 10 mg ml⁻¹ proteinase K were added. Cells were incubated at 37 °C for 1 h to allow complete lysis. 100 μl of 5 M NaCl was added to each tube and mixed thoroughly, before addition of 80 μl of 1% CTAB in 100 mM NaCl. The cell lysates were incubated at 65 °C for 10 min. Nucleic acids were purified by addition of an equal volume of Phenol:Chloroform: Isoamyl Alcohol (25:24:1, pH 8.0), vigorous mixing by inversion and centrifugation for 5 min at $14,000 \times g$. The upper aqueous layer containing DNA was carefully pipetted into a fresh tube and the phenol:chloroform:isoamyl alcohol step was repeated a further two times. Traces of phenol were removed by addition of an equal volume of chloroform, vigorous mixing by inversion and centrifugation for 5 min at $14,000 \times g$. The aqueous fraction was transferred to a fresh tube and nucleic acids were precipitated by addition of 0.6 volume of ice cold isopropanol, incubation at −20 °C for 1 h and centrifugation at $14,000 \times g$ for 20 min. DNA pellets were washed with 70% ethanol, air dried for ~10 min and resuspended in 50–100 μl of TE buffer. RNA was removed by addition of 1 μl of 10 mg ml⁻¹ RNase and incubation at 37 °C for 1 h. Total RNA was purified using the NucleoSpin RNA Kit (Macherey-Nagel) and DNAseI treated on column according to the recommended protocol. RNA was quantified using a Nanodrop spectro-photometer. 1 μg of total RNA was converted to cDNA using the LunaScript RT SuperMix Kit (NEB).

**RNAseq**. Production of GTAs is thought to lead to cell death through packaging of host cell's entire genome followed by lysis from within[18,19,42]. To inhibit lysis, cultures were grown in a high phosphate medium, RCV, to stationary phase where total RNA was isolated[18]. RNA yield was quantified and quality checked using a Nanodrop spectrophotometer and Aglient bioanalyser. Ribosomal RNA was removed from 1 μg good quality total RNA using the Ribo-Zero rRNA Removal Kit (Bacteria; Illumina). Libraries were then prepared from rRNA-depleted samples using the NEBNext RNA Ultra II Directional Library preparation kit for Illumina, with single 6 bp indices, according to the manufac-turer's guidelines for insert sizes of approximately 200–350 bp. Libraries were pooled at equimolar ratios, and the pool was sent for $2 \times 150$ base paired end sequencing on a HiSeq 3000 at the University of Leeds Next Generation Sequen-cing Facility.

Abundance of transcripts were compared between the wild-type *R. capsulatus* strain SB1003 ($n = 4$), a GTA hyperproducer DE442 ($n = 4$) and a DE442 culture that had been passaged three times ($n = 4$). Reads were quality checked and trimmed using FastQC version 11.0.5[43] and Cutadapt version 1.8.3[44], respectively. Kallisto version 0.43.1[45] was used to pseudo-align reads to the *R. capsulatus* SB1003 reference transcriptome, and to quantify gene expression. Differential expression analysis was performed using Sleuth version 0.29.0[46]. A full linear model containing strain, passage and sequencing batch was fit to the data. In order to look at the effect of strain, the full model was compared to a reduced model based only on passage and batch. The effect size of the test variable, i.e. strain DE442 vs

SB1003, was calculated using the Wald test to give the beta value (b), based on fitting a linear model to the data, in log2 units. The se_b value is the standard error. The q-value (qval) is the p-value adjusted by false discovery rate, where the p-value was calculated using the likelihood ratio test (LRT) in Sleuth. RNAseq data was submitted to the GEO database with the record ID GSE118116 - Comparison of the expression profiles of wild-type *Rhodobacter capsulatus* and a GTA hyperproducer (DE442) by RNAseq.

**Gene knockouts**. Knockouts were created by RcGTA transfer. pCM66T plasmid constructs were created with a gentamicin resistance cassette flanked by 500–1000 bp of DNA from either side of the target gene. Assembly was achieved by a one-step, four component NEBuilder (NEB) reaction and transformation into NEB 10-beta cells. Deletion constructs were introduced into the RcGTA hyperproducer strain by conjugation and a standard GTA bio-assay was carried out to replace the intact chromosomal gene with the deleted version.

**GafA Overexpression in *Rhodobacter***. Gene overexpression in *Rhodobacter* was achieved by a transcriptional fusion of the genes of interest to the *puf* photo-synthesis promoter[19]. Growth and general strain maintenance of *Rhodobacter* strains containing overexpression plasmids was carried out at 30 °C under aerobic, chemotrophic growth conditions where transcription from the *puf* promoter is strongly repressed. To produce overexpression conditions 12 ml cultures were grown to stationary phase aerobically, mixed 1:1 with fresh media and immediately transferred to 23 ml sealed tubes. Cultures were then incubated at 30 °C with illumination to induce *puf* promoter activity.

***Rhodobacter* gene transfer assays**. In *Rhodobacter*, the assays were carried out essentially as defined by Leung and Beatty (2013)[47]. RcGTA donor cultures were grown anaerobically with illumination in YPS for ~72 h and recipient cultures were grown aerobically in RCV for ~24 h. For overexpression experiments, donor cultures were first grown aerobically to stationary phase then anaerobically for 6 h or 24 h. Cells were cleared from donor cultures by centrifugation and the supernatant filtered through a 0.45 μm syringe filter. Recipient cells were concentrated 3-fold by cen-trifugation at $5000 \times g$ for 5 min and resuspension in 1/3 volume G-Buffer (10 mM Tris-HCl (pH 7.8), 1 mM MgCl₂, 1 mM CaCl₂, 1 mM NaCl, 0.5 mg ml⁻¹ BSA). Reactions were carried out in polystyrene culture tubes (Starlab) containing 400 μl G-Buffer, 100 μl recipient cells and 100 μl filter donor supernatant, then incubated at 30 °C for 1 h. A 900 μl volume of YPS was added to each tube and incubated for a further 3 h. Cells were harvested by centrifugation at $5000 \times g$ and plated on YPS + 100 μg ml⁻¹ rifampicin (for standard GTA assays) or 3 μg ml⁻¹ gentamicin (for gene knockouts).

**Quantitative reverse transcriptase PCR**. For each cDNA template, a 50-fold dilution was prepared in distilled water and 1 μl of diluted template was used per reaction. Reactions contained Fast Sybr Green Mastermix (Applied Biosys-tems), cDNA and primers (500 nM). Standard conditions were used with an annealing temperature of 60 °C. All primer efficiencies were calculated as between 90 and 110%. Relative gene expression was determined using the ΔΔCt method[48]. For each sample, variance was calculated for three independent biolo-gical replicates, which were each the mean of three technical replicates. Quant-Studio 3 Real-Time PCR System was used for all experiments (Applied Biosystems).

**Protein purification**. For His6-tagged proteins, 500 ml cultures of *E. coli* con-taining the relevant expression plasmid were induced at mid-exponential growth phase with 0.2 mM IPTG overnight at 20 °C. Concentrated cells were lysed in 20 ml binding buffer (1 M NaCl, 75 mM Tris; pH 7.75) plus 0.2 mg ml⁻¹ lysozyme and 500 U Basemuncher Endonuclease (Expedeon Ltd.) for 30 min on ice and then sonicated. Cleared supernatant was applied to a 5 ml HisTrap FF crude column (GE Healthcare) and the bound, his-tagged protein was eluted with 125 mM imidazole. Eluted protein was desalted on a HiPrep 26/10 desalting column (GE Healthcare) and then further separated by size exclusion chromatography on a HiLoad 16/60 Superdex 200 preparative grade gel filtration column. All chroma-tography steps were carried out on an AKTA Prime instrument (GE Healthcare). Purified proteins were concentrated in a Spin-X UF Centrifugal Concentrator (Corning) and quantified by the nanodrop extinction co-efficient method (Thermo Scientific). Samples were stored at −80 °C in binding buffer plus 50% glycerol. MBP-tagged proteins were purified as above except the cells were induced with 1 mM IPTG, MBP binding buffer was used (200 mM NaCl, 20 mM Tris, 1 mM EDTA; pH 7.4), the lysate was applied to a 5 ml MBPTrap FF column (GE Healthcare) and purified protein was eluted with 10 mM maltose in MBP binding buffer.

**Electrophoretic motility shift assays (EMSA)**. For all 50 bp binding substrates, 50 base Cy5 5′-labelled oligos (IDT) were annealed to unlabelled complimentary oligos (IDT). Both oligos were mixed to a final concentration of 40 μM in annealing buffer (1 M Potassium Acetate, 300 mM HEPES; pH 7.5) and heated to 98 °C for 5 min then allowed to cool to room temperature. 10 μl EMSA mixtures contained

80 nM annealed Cy5-dsDNA, GntR DNA binding buffer (25 mM HEPES, 50 mM K-glutamate, 50 mM MgSO$_4$, 1 mM dithiothreitol, 0.1 mM EDTA, 0.05% Triton X-100; pH 8.0)[49] for all assays except those testing GtaR for which a modification of the published GtaR binding buffer was used (10 mM HEPES, 40 mM NaCl; pH 8)[16], 1 μg poly dI:dC, 4% glycerol and the specified concentrations of purified protein[50]. 500-fold excess of competitor DNA was added to control mixtures – specific competitor was unlabelled but otherwise identical to the binding substrate and the non-specific competitor was an unlabelled 50 bp annealed oligo matching an arbitrary location elsewhere in the *R. capsulatus* genome. All assays except GtaR were incubated for 15 min at 30 °C then immediately loaded onto a 5% Acrylamide gel (1× TBE) without loading dye. GtaR assays were incubated at 37 °C for 30 minutes[16]. Gels were run at 100 V for 1 h at room temperature in 1× TBE. Fluorescence was imaged using a Typhoon Biomolecular Imager (Amersham) and analysed using ImageQuant (Amersham) and FIJI[51] software. For the full length RcGTA promoter (pGTA), a 5′ Cy5-labelled oligo was used to create a 633 bp PCR product. The pGTA DNA was used under the same conditions as the annealed oligos, except the concentration was 2 ng μl$^{-1}$, reactions were run at 100 V for 4 h at 4 °C. Non-fluorescent reactions used 100 ng of unlabelled PCR products as binding substrates and were run on 1% high resolution MicroSieve 3:1 Agarose (Cambridge Reagents) in 1× TBE at 100 V for 2 h. Gels were stained with Sybr Safe (Invitrogen) and imaged on a GelDoc transilluminator (BioRad).

***Ruegeria/Roseovarius* gene transfer assays**. Assays were carried out as originally reported in Biers et al.[52]. In brief, spontaneous rifampicin or strepto-mycin resistant colonies were isolated by plating onto selective MB agar. Cultures were grown in ½YTSS medium for 5 days, static and without illumination. For co-culture experiments, a rifampicin resistant strain was grown together with a streptomycin resistant strain then plated on marine broth agar with both anti-biotics to assess transfer of resistance. For in vitro assays, resistant strains were grown separately for 5 days and filtered through a 0.45 μm syringe filter. The filtered supernatant was then added to antibiotic sensitive cells, shaken at 200 rpm for 1 h in the dark and plated on marine broth agar containing the relevant antibiotics. The *gafA* homologues were cloned into pSRKBB to produce pCMF195 & 6 (Supplementary Table 5); *gafA* expression was induced from the lac′ promoter by addition of 1 mM IPTG when growth had reached late logarithmic phase (OD$_{600}$: ~0.8–1.0).

**Bioinformatics**. Helix turn helix predictions were carried out using NPS@[53,54] and Gym2.0[55] using the default settings. HHPRED[56,57] analysis of GafA was carried out using the pdb_mmcif70_5_oct database and the default parameters i.e. HHBlits uniprot20_2016_02 MSA generation method, maximal generation steps = 3 and an E-value threshold of 1e-3. Minimum coverage was 20%, minimum sequence identity was 0%. Secondary structure scoring was done during alignment (local). Initial full length protein query was refined and resubmitted according to the automatic suggestions provided by the software for the two respective domains. The NCBI BlastP search for GafA homologues was performed with the default parameters - expect threshold = 10, word size = 6, blosum62 similarity matrix, gap costs of existence = 11 and extension = 1. No taxonomic constraints were applied but sequences from uncultured/environmental samples were excluded. The top ten hits belonging to different species were arbitrarily selected for analysis irrespective of alignment score, the most distant match used (*Sulfitobacter spp.*) produced a score of 377 and an E value of 6e-126 from 100% coverage and 55% sequence identity. Promoter sequences for each protein were then identified in the nucleotide database for each sequence. Promoter −10/−35 elements were predicted with BPROM[58]. FIJI software[51] was used to measure band intensities in EMSA experiments with the Gel Analyzer plug in, ClustalW2[59] and ClustalΩ[60] were used for DNA/protein alignments as indicated in the figure legends, Jalview[61] was used to visualize alignments. Transcript abundance was visualised using the Broad Institute's IGV viewer[62]. Statistical analysis was carried out using Sigmaplot soft-ware version 13 (Systat Software Inc., www.systatsoftware.com.) and, for each use, the test parameters are indicated in the text and/or figure legends. The Sigmaplot Ligand Binding macro was also used to calculate dissociation constants (kD) from EMSA band intensities.

**Reporting Summary**. Further information on experimental design is available in the Nature Research Reporting Summary linked to this article.

## Data availability

All the data needed to evaluate the conclusions of the paper are present in the paper and the Supplementary Information files. Source data for all graphs and gel images are provided as a Source Data file. The complete RNAseq data was submitted to the NCBI Gene Expression Omnibus (GEO) Database, accession number GSE118116 [https://www.ncbi.nlm.nih.gov/geo/query/acc.cgi?acc=GSE118116].

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

## Acknowledgements

The author thanks the University of York Technology Facility for providing access to equipment and expert technical assistance when required, with particular acknowledgement to Dr. Katherine Newling for RNAseq quality control and statistical analysis. The author also thanks Dr. Jelena Kusakina for critical reading of the manuscript. This work was wholly supported by a Wellcome Trust/Royal Society Sir Henry Dale Fellowship Grant (109363/Z/15/Z).

## Author contributions

P.C.M.F. conceived, designed and implemented this study and prepared the manuscript.

## Additional information

**Competing interests:** The author declares no competing interests.

