## [Peer Review File · Nature Communications]

Reviewers' comments:

Reviewer #1 (Remarks to the Author):

Review of Paul Fogg ms for Nature Communications.

Gene Transfer Agents (GTAs) are exciting and enigmatic virus-like particles produced by some Bacteria and Archaea for yet undetermined purpose. In the past decade, many genes involved in GTA production were identified, and especially for GTAs formed by a bacterium *Rhodobacter capsulatus* (RcGTA). A few of these genes remain either partially characterized or completely uncharacterized, hindering our understanding of RcGTA production, functionality, and evolution. In 2016, an additional gene (*Rhodobacter capsulatus* locus tag *rcc01865*) was identified as essential for the production of RcGTA particles (Hynes et al 2016; ref. 22 in the manuscript). Based on the experimental and bioinformatic analyses, the gene product was predicted to interact with DNA and hypothesized to be involved in gene regulation (Hynes et al 2016). In this manuscript, Paul Fogg and colleagues provide experimental evidence that the *rcc01865* gene product is indeed involved in gene regulation, and more specifically serves as a direct RcGTA activator in *R. capsulatus*, and confirm that this gene has the same function in closely related GTA producers *Ruegeria mobilis* and *Roseovarius nubinhibens*. The obtained data allowed authors to introduce a detailed model of RcGTA regulation, which is an important advance in our understanding of GTA production in *R. capsulatus*.

The manuscript, however, has some organizational issues, lacks descriptions of a few methods and results (especially those pertaining to computational sequence analyses) and has a few places where clarifications need to be made. The manuscript also sometimes present as novel the observations that were already reported in earlier studies. These issues are detailed in the comments below and should be corrected.

Specific major comments:

Abstract and Introduction:

1. Lines 3-18: The abstract presents GafA as a gene that work in all GTAs, which is not what was demonstrated. There are several GTAs unrelated to RcGTA. The manuscript mostly deals with RcGTA and its counterparts in two closely related bacterial species. The abstract should be revised to reflect this.
2. Line 9, "in any bacterial species": The manuscript shows this only for three species, not for "any species". Plus, the *rcc01865* homologs are undetectable outside of Rhosobacterales and

Rhizobiales, while other RcGTA-like genes are found in many other alpha-proteobacterial taxa (see Shakya et al 2017 for details; PMID 29250433).

3. Lines 19-26: Discussion of infectious diseases and HGT of antibiotic resistance seems irrelevant to this manuscript, as *Rhodobacter capsulatus* and other two species are not human pathogens, and to my knowledge there is no evidence that antibiotic resistance genes are transferred via GTAs (other than in the laboratory constructs).

4. Lines 27-28, “a forth mechanism”: There are actually more than 3 mechanisms of HGT. More pertinently, GTAs are indisputably related to phages, so GTA-mediated transfer is a form of transduction. The authors themselves refer to GTA-mediated transfer as “transduction” later in the manuscript.

Results:

5. “GTA” should be instead “RcGTA” in almost all instances throughout the Results, as the data shown relates mostly to RcGTA and not to GTAs in general.

6. Line 46, “The top 28 upregulated genes...”: What was the criterion for selecting the top 28 genes? If a value of beta > 4 was used, there should be 29 genes (based on data shown in the Additional Table 1). Regardless, the criterion should be provided either in the Results or in Methods.

7. Line 48, “Of the remaining five genes...”: There are 24 known genes associated with RcGTA production (Hynes et al 2016), all of which are indeed in the top 28 genes, but then there are 28-4=4 genes left, not five. This comment is related to the previous comment – perhaps 28 should be 29? There is another mix up here: the *rcc01865* gene should already be included in the set of 24 genes, so it should not be counted towards the remaining 4 (or 5) genes.

8. Line 49, “its role is unknown”: not entirely. Hynes et al. 2016 showed that (a) *R. capsulatus rcc01865* mutant lacked RcGTA activity; (b) predicted that the encoded protein contains helix-turn-helix DNA binding motif, has many charged amino acids and consists predominantly of alpha helical secondary structure; (c) as a result, hypothesized the gene/protein role in RcGTA regulation. This information should be presented in this manuscript as already known and not as new analyses performed by the authors.

9. Line 54, “cluster together”: it is not clear what the authors mean here– there is no clustering analysis presented in Extended Data Table 3.

10. Line 60, “all stages of GTA production are inhibited”: perhaps, this is an overstatement? The assay in Fig. 1B only tests DNA packaging; therefore, at least hypothetically, capsids/particles may still be produced, but DNA was not packaged into them.

11. Line 62, “equivalent”: based on Fig. 1C, the transduction frequency is much lower in *gafAOX* in comparison to DE442 strain. However, the statistical comparison is not shown on the figure and not described in the text: was the difference not significant, or was the comparison not performed?

12. Line 79, “did not rescue GTA transduction ability”: Evidence for this is not shown on Figure 1F, which is referred to at the end of the sentence.
13. Line 118, “rearranged version”: Why is it referred as “rearranged”? I see in Figure 3A some nucleotide substitutions, but I do not see evidence of rearrangements.
14. Lines 147-160: Observations of the *gafA* presence in Rhodobacterales, Rhizobiales and RDJLphi1 phage reported here were already described in Hynes et al (2016) and Shakya et al (2017), and this should be attributed here. Also, RDJLphi1 phage contains only part of the *ctrA* homolog: this is alluded to in Figure S10 legend, but the manuscript narrative makes an impression of the presence of complete *ctrA* homolog. Lang et al (2017; PMID 28784044) investigate evolutionary history of *ctrA* homologs in RDJLphi1 and RDJLphi2 phages in detail.
15. Line 147-149, “GafA is conserved throughout Rhodobacteraceae [...] both in terms of primary sequence and genomic context (Extended Data Fig. 7 and 8)”: How do shown figures support this claim? Figure S7 shows alignment of only 3 Rhodobacteraceae *gafA* homologs. Figure S8 shows a relationship among selected *gafA* homologs, but that is not relevant to the statement of this sentence, as it does not summarize absence of the homologs among members of Rhodobacteraceae (if any). Additionally, I do not understand what the authors mean by “genomic context”. The sentence is also in disagreement with the reported observation of Hynes et al 2016, who have shown that the *rcc01865* gene is among the fastest evolving RCGTA-like genes in Rhodobacteraceae (and hence the gene is NOT very conserved on the primary sequence level). However, based on the 0.3 substitutions/site scale bar of Figure S8, I infer that the Rhodobacteraceae homologs on the presented phylogenetic tree are quite divergent, which is consistent with the observation of Hynes et al. (2016) and in disagreement with the authors’ interpretation.

Methods:

16. The description of RNA-Seq analysis performed after the sequencing step is insufficient. How was the quality control was performed? What software was used for the analysis? What is the test variable? Explain how beta was calculated in detail. What criteria were used to select the top 28 (or 29; see comment above) genes?
17. Specifics of the HHPRED analyses described in Figure S1 should be described in Methods, as the program has many options. E-value for the sigma-factor domain is high, but it is dependent on the database size, and it is not described what database was used to calculate it and what type of search was used.
18. Parameters of the similarity search performed to generate data on Figure S4 should be described. What was the E-value/bitscore cutoff? What scoring scheme was used? Were any taxonomic limits imposed? E-value depends on the database size, so access date to the nr database or release version should be included.

19. Phylogenetic analyses performed to produce tree shown on Figure S8 are not well described. What were the parameters in PSI-BLAST search? What substitution model was used in tree reconstruction? How was the substitution model selected? Was there any assessment of tree reconstruction accuracy (like bootstrapping; the support values should be shown on the tree)? Is tree unrooted? Also, why were Rhizobiales homologs excluded from the analyses?

Figures:

20. Panels of Figure 1 seem not to have a unifying theme to have them in one figure, and the figure legend is a bit disorganized.

21. Figure 1A, in my opinion, is uninformative, as the relevant numbers are explicitly discussed in the manuscript and presented in the Additional Data Table 1. I suggest panel A to be removed.

22. Figure 1F: Lanes 6 and 7 are unclearly labeled and insufficiently described. Do both lanes have the same label that consists of two lines? Do these two lanes represent two replicates?

Other, minor comments:

23. Lines 38-39: The connection between description of marine viromes and GTA production in *R. capsulatus* is unclear.

24. Line 58, "DE442": should be "*R. capsulatus* DE442", as this is the first mentioning of this strain in the text.

25. Line 63, "Fig. 2D" should be "Fig. 1D".

26. Line 97: "GTA production e.g." should be "GTA production, e.g.".

27. Line 138-139, "the known GtaR binding site": Reference 16 should be cited here.

28. Lines 141-142, "GTAs are normally produced in a small proportion of any given population...": citations to relevant studies are needed to be provided for this sentence.

29. Data availability: please provide GEO accession numbers. Also, will the processed RNA-seq data be available via public data repositories like DataDryad or FigShare?

30. Figure 2: Explain in the figure legend what "delta-delta-Ct" abbreviation on Y axes designate.

31. Figure 2 panels are organized in a confusing order: 2E is referred to earlier than 2C or 2D.

32. It is confusing that table captions appear after the tables themselves.

33. Table 1: provide locus tags for all genes, even if they have gene names like "rpsR". This will help readers to identify genes in question faster.

34. Table 3 caption: what are the "test variable" units? Please explain here (as well as in Methods; see earlier comment.)

35. Sometimes supplementary figures and tables are referred with a prefix "S", and other times as "Additional Data Figure/Table".

36. Figure S7 legend: BLOSUM62 is a substitution matrix and not an algorithm. Could the author elaborate how the BLOSUM62 matrix was used for similarity shading, as such procedure, to my knowledge, is not the default coloring scheme produced by the ClustalW program (i.e., as defined in the colprot.xml file that comes with the program)?

37. Figure S7 legend: What version of ClustalW program was used? A reference to the paper describing the program should be provided.

Reviewer #2 (Remarks to the Author):

In this manuscript the author characterised GafA as an activator of the gene transfer agents (GTAs). The author has also performed an initial analysis about how this regulator interacts with others in this process. While GTAs are very interesting elements promoting gene transfer, and the characterisation of this activator may be important for the people working on this field, the current manuscript does not provide a big picture of the system: i.e. it is not clear when this inducer is expressed or what are the signals that promote its expression. In its current form, and in opinion of this reviewer, the manuscript does not provide sufficient insights into the system to be published in Nature Communications. Note that, as indicated in the text, the initial identification and characterisation of the GafA activator was performed previously (doi.org/10.1093/molbev/msw125), which severely impacts the novelty of this study. In that paper the authors already demonstrated the essentially of the gene, and proposed that this protein acts as a gene regulator (see Table 1).

Other comments:

- The results are difficult to follow. Too many results are presented in a single paragraph, so many times the rationale of the different experiments is difficult to understand. The results section should be expanded, and the rationale of the different experiments clearly explained.

- Not sure GTAs can be considered as the fourth mechanism of gene transfer. At the end, it resembles phage transduction.

- Line 33: the number of genes mobilised by transduction is also unlimited.

- Line 60: The lack of the 4 kb band in the *gafA* mutant does not indicate that all stages of GTA production are inhibited. Inhibition of any single step will also prevent the formation of this packaged band.

- Line 82: if the author's hypothesis about the role of CtrA in releasing the GTA particles is right, artificial lysis of the cells would restore GTA-mediated transfer.

- Line 84: The results presented here contradict the hypothesis presented in line 82. The fact that deletion of *ctrA* reduces expression of different genes indicates that *ctrA* is also involved (directly or indirectly) in the expression of the GTA structural genes.

- Since *crtA* induces *gafA* transcription, it is not clear for this reviewer why overexpression of the *crtA* gene does not increase significantly transcription of the GTA structural genes. Is this suggesting additional levels of control (i. e. post-transcriptional or translational modifications)?

- Some of the figures should include statistical analyses.

- To validate the experiments, the EMSA experiments should include competition with unlabelled DNA and competition with unrelated DNA.

- More importantly, the *ctrA* binding sites should be mutated and the impact of these mutations in terms of GTA transfer should be evaluated.

Reviewer #3 (Remarks to the Author):

This manuscript uses comparative transcriptomics to identify a likely key regulator of the life cycle of gene transfer agents (GTAs). GTAs represent a fourth mechanism of horizontal gene transfer agent in bacteria and are interesting from an evolutionary perspective in that they do not transfer their own DNA, but rather randomly package small segments of host DNA and transfer them to other members of the same species.

The manuscript identifies GafA as a direct activator of the GTA, and uses a number of techniques to investigate the relationship between GafA, CtrA and several other transcription factors known to be involved in GTA lifecycle. The manuscript concludes with a proposed model of GTA regulation, which attempts to bring together a number of observations from the literature including the roles of GtaR, CtrA, ClpXP, LexA and various phosphorelay components.

Statistics seem appropriate.

The results of this study appear to be novel and would clearly be of interest to others in the microbiology/evolution community.

Although the results are intriguing, I felt that stronger biochemical evidence for GafA binding, and some pursuit of identification of its binding site is required, as this is a major claim of the manuscript. A relatively large segment of DNA (633bp) is used in the band shift assay (Fig 3D). It isn't made clear (i) how well the *gafA* promoter sequence is defined, (ii) where the promoter is located within this 633 bp fragment, or (iii) why this particular sized fragment was chosen. Although an MBP-*gafA* fusion protein is used out of necessity, further experiments could be performed relatively easily here. Why not use progressively smaller DNA fragments in the band shift assay, or ideally use DNase footprinting to more closely define the GafA binding site? The most likely location for GafA binding would seem to be close to the promoter, as is the case for CtrA. It would be particularly interesting to see the relative positions of the CtrA and GafA (and GtaR); see below) binding sites at the pGafA promoter.

Two shifted bands are seen in the GafA band shift assay. With such a large fragment of DNA, can it be excluded that there are two independent sites on the fragment. Is there a biochemical explanation for the presence of two bands?

Similarly, a useful experiment to provide stronger evidence for GafA as a key regulatory protein might be to make a mutation at one or two key residues within the HTH motif (mutating to a residue that is often found at this position) and see if binding is lost in the band shift assay.

The CtrA band shift assays (Fig 3B) use a 50bp fragment and two shifted bands are annotated on the figure. Again is there biochemical explanation? There is a third band (lower mobility) which appears in the first three lanes with low concentrations of *ctrA* present. Is this significant? – whether this band is also present in the no protein control lane is a little hard to see due to bleed through on the left.

Some discussion of the role of sigma factors in GTAs would be appropriate. Mercer et al (2014) (ref 23 in the manuscript) explored the effects of knocking out of each of the non-essential host sigma

factors. No impact on GTA was observed. Given the rpoE domain found in GafA, could GafA act as an alternative sigma factor?

The results will no doubt influence thinking in this field. A paper as recent as June 2018

(Appl Environ Microbiol 84:e00275-18. <https://doi.org/10.1128/AEM.00275-18>) states that the transcription factors acting on GTA promoters remain a mystery, indicating the work here is important. I note that those authors also state that they saw no CtrA binding around GTA promoters and refer to unpublished band shift assays. This is in contrast to the strong binding reported in the present manuscript. Hence I feel there is a need to present the best evidence possible for specific DNA binding in the manuscript here.

Figure legends could be made more informative. For example, to specify what size fragments are used for the gel shifts – partially described in methods but some use short oligos, other use much longer (633bp) DNA sequences, others eg endolysin promoter (fig S5B) not described.

Panels often seem to be presented out of order in the legends

Fig 3A – It would be helpful for the reader to show other promoter features here, if known (eg RNAP binding sequences).

Fig. S1 – The meaning of the HTH alignment below the HTH motif wasn't clear.

Fig. S6. Predicted GtaR binding sites at the endolysin promoter and gafA promoter. Indicate where these are in the context of the promoter. What is the position of the proposed GtaR site at pGafA relative to the CtrA binding site shown in Fig S4? Given GtaR is a key protein in GTA production (Fig 4) the author might also consider providing direct in vitro evidence (band shift/footprint) for GtaR binding at the proposed site.

Reviewers' comments:

Reviewer #1 (Remarks to the Author):

Review of Paul Fogg ms for Nature Communications.

Gene Transfer Agents (GTAs) are exciting and enigmatic virus-like particles produced by some Bacteria and Archaea for yet undetermined purpose. In the past decade, many genes involved in GTA production were identified, and especially for GTAs formed by a bacterium *Rhodobacter capsulatus* (RcGTA). A few of these genes remain either partially characterized or completely uncharacterized, hindering our understanding of RcGTA production, functionality, and evolution. In 2016, an additional gene (*Rhodobacter capsulatus* locus tag rcc01865) was identified as essential for the production of RcGTA particles (Hynes et al 2016; ref. 22 in the manuscript). Based on the experimental and bioinformatic analyses, the gene product was predicted to interact with DNA and hypothesized to be involved in gene regulation (Hynes et al 2016). In this manuscript, Paul Fogg and colleagues provide experimental evidence that the rcc01865 gene product is indeed involved in gene regulation, and more specifically serves as a direct RcGTA activator in *R. capsulatus*, and confirm that this gene has the same function in closely related GTA producers *Ruegeria mobilis* and *Roseovarius nubinhibens*. The obtained data allowed authors to introduce a detailed model of RcGTA regulation, which is an important advance in our understanding of GTA production in *R. capsulatus*.

The manuscript, however, has some organizational issues, lacks descriptions of a few methods and results (especially those pertaining to computational sequence analyses) and has a few places where clarifications need to be made. The manuscript also sometimes present as novel the observations that were already reported in earlier studies. These issues are detailed in the comments below and should be corrected.

Specific major comments:

Abstract and Introduction:

1. Lines 3-18: The abstract presents GafA as a gene that work in all GTAs, which is not what was demonstrated. There are several GTAs unrelated to RcGTA. The manuscript mostly deals with RcGTA and its counterparts in two closely related bacterial species. The abstract should be revised to reflect this.

It was not the intention of the abstract to imply that GafA works in all GTAs, but rather that it works as an activator in the species tested that have RcGTA-like GTAs and it is possible that future work will allow identification of similar activators in other GTA producers. The text has been revised to reflect this.

2. Line 9, "in any bacterial species": The manuscript shows this only for three species, not for "any species". Plus, the rcc01865 homologs are undetectable outside of Rhosobacterales and Rhizobiales, while other RcGTA-like genes are found in many other alpha-proteobacterial taxa (see Shakya et al 2017 for details; PMID 29250433).

This is a misunderstanding. By “in any species” the meaning was not that GafA is present/active in all species but rather it is the first instance of a direct GTA activator of any kind to be reported for any GTA. The text has been updated to clarify.

3. Lines 19-26: Discussion of infectious diseases and HGT of antibiotic resistance seems irrelevant to this manuscript, as *Rhodobacter capsulatus* and other two species are not human pathogens, and to my knowledge there is no evidence that antibiotic resistance genes are transferred via GTAs (other than in the laboratory constructs).

It is true that transfer of antibiotic resistance by GTAs in the environment has not been assessed before, probably because natural transfer would be indistinguishable from other methods of acquisition when looked at post factum. However, McDaniel et al 2010 did look at mobilization of two antibiotic resistance genes in an in situ marine study and found high frequency gene transfer even cross genera in some cases. I agree that the species looked at here, which are model systems, are not pathogens but it is increasingly accepted that environmental reservoirs of antibiotic resistance are an important source of resistance transfer to pathogens e.g. (Forsberg et al., 2012). Furthermore, GTAs are found in animal and plant pathogens and future work in our lab will explore these species in more detail. The text has been updated to clarify but my preference is to retain the general point.

4. Lines 27-28, “a forth mechanism”: There are actually more than 3 mechanisms of HGT. More pertinently, GTAs are indisputably related to phages, so GTA-mediated transfer is a form of transduction. The authors themselves refer to GTA-mediated transfer as “transduction” later in the manuscript.

The 3 mechanisms of HGT referred to are the three traditional major classes i.e. transduction, conjugation and transformation (Davison, 1999), of course there are sub divisions but these three umbrella terms are well established. As the reviewer states, GTAs are undoubtedly related to phages but there are a number of fundamental differences that prevent them from being classed as phages. GTAs are essentially a hybrid of phage transduction and natural transformation, the donor cell produces GTAs in a manner similar to phage (except there is no preference for the GTAs own genes) but the recipient takes up the DNA and incorporates it into its own genome using the competence pathway and homologous recombination (Brimacombe et al., 2015). The text has been revised to reflect this and all references to GTA transduction have been changed to GTA gene transfer.

Results:

5. “GTA” should be instead “RcGTA” in almost all instances throughout the Results, as the data shown relates mostly to RcGTA and not to GTAs in general.

Agreed and corrected

6. Line 46, “The top 28 upregulated genes...”: What was the criterion for selecting the top 28 genes? If a value of $\beta > 4$ was used, there should be 29 genes (based on data shown in the Additional Table 1). Regardless, the criterion should be provided either in the Results or in Methods.

Agreed and corrected

7. Line 48, “Of the remaining five genes...”: There are 24 known genes associated with RcGTA production (Hynes et al 2016), all of which are indeed in the top 28 genes, but then there are 28-4=4 genes left, not five. This comment is related to the previous comment – perhaps 28 should be 29?

There is another mix up here: the rcc01865 gene should already be included in the set of 24 genes, so it should not be counted towards the remaining 4 (or 5) genes.

Agreed and corrected

8. Line 49, “its role is unknown”: not entirely. Hynes et al. 2016 showed that (a) *R. capsulatus* rcc01865 mutant lacked RcGTA activity; (b) predicted that the encoded protein contains helix-turn-helix DNA binding motif, has many charged amino acids and consists predominantly of alpha helical secondary structure; (c) as a result, hypothesized the gene/protein role in RcGTA regulation. This information should be presented in this manuscript as already known and not as new analyses performed by the authors.

Text modified to clarify. Although Hynes et al. found that a gene knock-out lacked RcGTA activity, the precise function was unclear. There are several genes that reduce, ablate or modify RcGTA function that do not act directly e.g. CtrA, GtaR/I, cckA. The sequence analysis by itself was not sufficient to assign definitive function.

9. Line 54, “cluster together”: it is not clear what the authors mean here— there is no clustering analysis presented in Extended Data Table 3.

The text has been updated to clarify

10. Line 60, “all stages of GTA production are inhibited”: perhaps, this is an overstatement? The assay in Fig. 1B only tests DNA packaging; therefore, at least hypothetically, capsids/particles may still be produced, but DNA was not packaged into them.

The text has been updated to clarify. When take together with the qPCR data in this paper, I believe that expression of the core RcGTA cluster is silenced.

11. Line 62, “equivalent”: based on Fig. 1C, the transduction frequency is much lower in gafAOX in comparison to DE442 strain. However, the statistical comparison is not shown on the figure and not described in the text: was the difference not significant, or was the comparison not performed?

The text has been updated to clarify. It was tested and it was significant, Figure amended.

12. Line 79, “did not rescue GTA transduction ability”: Evidence for this is not shown on Figure 1F, which is referred to at the end of the sentence.

Figure 1F is to illustrate the second half of the sentence i.e. that DNA packaging is restored. The lack of transduction restoration was not presented because it is essentially an empty graph i.e. ctrA knock-out is 0% and ctrA ko + gafA is also 0%. Sentence updated to clarify.

13. Line 118, “rearranged version”: Why is it referred as “rearranged”? I see in Figure 3A some nucleotide substitutions, but I do not see evidence of rearrangements.

Text updated to clarify. Essentially, based on previous work in Caulobacter, the CtrA-binding site in the CtrA promoter is TTAA-N7-TTAA. Here, we have either GTAAC-N6-TTAAC or TTAAC-N6-GTAAC where the two half site versions are swapped.

14. Lines 147-160: Observations of the gafA presence in Rhodobacterales, Rhizobiales and RDJLphi1 phage reported here were already described in Hynes et al (2016) and Shakya et al (2017), and this should be attributed here. Also, RDJLphi1 phage contains only part of the ctrA homolog: this is alluded to in Figure S10 legend, but the manuscript narrative makes an impression of the presence of complete ctrA homolog. Lang et al (2017; PMID 28784044) investigate evolutionary history of ctrA homologs in RDJLphi1 and RDJLphi2 phages in detail.

Text updated to clarify.

15. Line 147-149, “GafA is conserved throughout Rhodobacteraceae [...] both in terms of primary sequence and genomic context (Extended Data Fig. 7 and 8)”: How do shown figures support this claim? Figure S7 shows alignment of only 3 Rhodobacterceae gafA homologs. Figure S8 shows a relationship among selected gafA homologs, but that is not relevant to the statement of this sentence, as it does not summarize absence of the homologs among members of Rhodobacteraceae (if any). Additionally, I do not understand what the authors mean by “genomic context”. The sentence is also in disagreement with the reported observation of Hynes et al 2016, who have shown that the rcc01865 gene is among the fastest evolving RcGTA-like genes in Rhodobacteraceae (and hence the gene is NOT very conserved on the primary sequence level). However, based on the 0.3 substitutions/site scale bar of Figure S8, I infer that the Rhodobacteraceae homologs on the presented phylogenetic tree are quite divergent, which is consistent with the observation of Hynes et al. (2016) and in disagreement with the authors’ interpretation.

Although Hynes et al reported that 1865 was among the 6 most rapidly evolving GTA genes, the difference between 1865 and many of the other GTA genes was not great. In general their findings were that GTA genes are evolving slightly faster than core host genes but much slower than comparable phage genes, there were insufficient 1865 homologues in phage for comparison but the PPD figures reported were ~1.5 for 1865 compared to ~1 for other GTA genes and >3 for phage genes. Text updated to clarify that GafA (1865) is present, rather than conserved, throughout the Rhodobacterales and reference is made to previous work noting the increased evolution of this gene compared to most RcGTA genes. Extended Data figure 8 has been deleted and replaced with discussion of previous work.

Methods:

16. The description of RNA-Seq analysis performed after the sequencing step is insufficient. How was the quality control was performed? What software was used for the analysis? What is the test variable?

Text expanded

Explain how beta was calculated in detail. What criteria were used to select the top 28 (or 29; see comment above) genes?

Done

17. Specifics of the HHPRED analyses described in Figure S1 should be described in Methods, as the program has many options. E-value for the sigma-factor domain is high, but it is dependent on the database size, and it is not described what database was used to calculate it and what type of search was used.

The figure has been updated using the current PDB_mmCIF70_5_Oct database and the annotations reflect this. More detail has been added in the methods section.

18. Parameters of the similarity search performed to generate data on Figure S4 should be described. What was the E-value/bitscore cutoff? What scoring scheme was used? Were any taxonomic limits imposed? E-value depends on the database size, so access date to the nr database or release version should be included.

Blast parameters were the default: Expect threshold 10, word size 6, blosum62, gap costs existence 11 extension 1. No taxonomic restrictions were applied. Cut off was not relevant, the

top 10 was arbitrary with the most distant match used (Sulfitobacter) producing a score of 377, 100% coverage, E= 6e-126 and Identity was 55%. Ruegeria, Roseovarius, Dinoroseobacter all added independently (i.e. without a Blast search) as known GTA producers. Access date 14th June 2018. Additional text added to the manuscript.

19. Phylogenetic analyses performed to produce tree shown on Figure S8 are not well described. What were the parameters in PSI-BLAST search? What substitution model was used in tree reconstruction? How was the substitution model selected? Was there any assessment of tree reconstruction accuracy (like bootstrapping; the support values should be shown on the tree)? Is tree unrooted? Also, why were Rhizobiales homologs excluded from the analyses?

This figure has now been deleted. Rhizobiales were not intentionally excluded but were more distantly related than the sequences presented and are also often split into two separate ORFs, complicating analysis.

Figures:

20. Panels of Figure 1 seem not to have a unifying theme to have them in one figure, and the figure legend is a bit disorganized.

Figure 1 has been modified

21. Figure 1A, in my opinion, is uninformative, as the relevant numbers are explicitly discussed in the manuscript and presented in the Additional Data Table 1. I suggest panel A to be removed.

Deleted

22. Figure 1F: Lanes 6 and 7 are unclearly labeled and insufficiently described. Do both lanes have the same label that consists of two lines? Do these two lanes represent two replicates?

Yes they were duplicates. Figure has been amended for clarity.

Other, minor comments:

23. Lines 38-39: The connection between description of marine viromes and GTA production in *R. capsulatus* is unclear.

The virome DNA that is unrelated to known virus genes is assumed to come from thus far uncharacterized viruses but there has been speculation that because GTAs package random DNA they could be a significant contributor to this “non-viral DNA” (e.g. Kristensen et al 2010)

24. Line 58, “DE442”: should be “*R. capsulatus* DE442”, as this is the first mentioning of this strain in the text.

Amended

25. Line 63, “Fig. 2D” should be “Fig. 1D”.

Amended

26. Line 97: “GTA production e.g.” should be “GTA production, e.g.”.

Amended

27. Line 138-139, “the known GtaR binding site”: Reference 16 should be cited here.

Amended

28. Lines 141-142, “GTAs are normally produced in a small proportion of any given population...”: citations to relevant studies are needed to be provided for this sentence.

Amended

29. Data availability: please provide GEO accession numbers. Also, will the processed RNA-seq data be available via public data repositories like DataDryad or FigShare?

This was not available at initial submission, accession is now provided.

30. Figure 2: Explain in the figure legend what “delta-delta-Ct” abbreviation on Y axes designate.

Amended

31. Figure 2 panels are organized in a confusing order: 2E is referred to earlier than 2C or 2D.

Amended

32. It is confusing that table captions appear after the tables themselves.

Amended

33. Table 1: provide locus tags for all genes, even if they have gene names like “rpsR”. This will help readers to identify genes in question faster.

34. Table 3 caption: what are the “test variable” units? Please explain here (as well as in Methods; see earlier comment.)

35. Sometimes supplementary figures and tables are referred with a prefix “S”, and other times as “Additional Data Figure/Table”.

Amended

36. Figure S7 legend: BLOSUM62 is a substitution matrix and not an algorithm. Could the author elaborate how the BLOSUM62 matrix was used for similarity shading, as such procedure, to my knowledge, is not the default coloring scheme produced by the ClustalW program (i.e., as defined in the colprot.xml file that comes with the program)?

Figure updated.

37. Figure S7 legend: What version of ClustalW program was used? A reference to the paper describing the program should be provided.

Figure updated.

Reviewer #2 (Remarks to the Author):

In this manuscript the author characterised GafA as an activator of the gene transfer agents (GTAs). The author has also performed an initial analysis about how this regulator interacts with others in this process. While GTAs are very interesting elements promoting gene transfer, and the characterisation of this activator may be important for the people working on this field, the current manuscript does not provide a big picture of the system: i.e. it is not clear when this inducer is expressed or what are the signals that promote its expression. In its current form, and in opinion of this reviewer, the manuscript does not provide sufficient insights into the system to be published in Nature Communications. Note that, as indicated in the text, the initial identification and characterisation of the GafA activator was performed previously (doi.org/10.1093/molbev/msw125), which severely impacts the novelty of this study. In that paper the authors already demonstrated the essentially of the gene, and proposed that this protein acts as a gene regulator (see Table 1).

The paper referred to here (Hynes et al 2016) does indeed identify gafA (rcc01865) as an essential gene for RcGTA production, however, a number of other genes have previously been identified that are required for RcGTA production but do not act directly, e.g. CtrA, CckA, GtaR, and as a

consequence all models of RcGTA regulation are fragmented. Hynes' proposal that the protein acts as a gene regulator was speculative (i.e. bioinformatic detection of a HTH and the fact it wasn't detected in a Chen's structural GTA proteome study) and was not backed up by any experimental data. Here, definitive data is presented that GafA is the first transcription factor to directly bind to the RcGTA promoter and to induce RcGTA production, it is involved in the lytic process together with CtrA and its promoter is in turn under the control of both CtrA and GtaR (quorum sensing). The model proposed here is the first to link the various known global RcGTA regulators together via GafA and thus is a major advance in our understanding of GTA biology.

Other comments:

- The results are difficult to follow. Too many results are presented in a single paragraph, so many times the rationale of the different experiments is difficult to understand. The results section should be expanded, and the rationale of the different experiments clearly explained.

Text has been updated

- Not sure GTAs can be considered as the fourth mechanism of gene transfer. At the end, it resembles phage transduction.

As stated earlier – "GTAs are undoubtedly related to phages but there are a number of fundamental differences that prevent them from being classed as phages. GTAs are essentially a hybrid of phage transduction and natural transformation, the donor cell produces GTAs in a manner similar to phage (except there is no preference for the GTAs own genes) but the recipient takes up the DNA and incorporates it into its own genome using the competence pathway and homologous recombination (Brimacombe et al., 2015). The text has been revised to reflect this."

- Line 33: the number of genes mobilised by transduction is also unlimited.

This is true but far less frequent that for GTAs. Transduction of non-viral genes by phages is usually either limited to those in close proximity to the integration site of the virus or, in the case of generalized transducing phage, host DNA is not the main target of the virus and make up only a small percentage of the packaged DNA. GTAs, on the other hand, exclusively package host DNA that always covers the whole genome and in the case of RcGTA and others there is no detectable bias. Text has been amended to clarify the distinction.

- Line 60: The lack of the 4 kb band in the gafA mutant does not indicate that all stages of GTA production are inhibited. Inhibition of any single step will also prevent the formation of this packaged band.

Amended

- Line 82: if the author's hypothesis about the role of CtrA in releasing the GTA particles is right, artificial lysis of the cells would restore GTA-mediated transfer.

The hypothesis proposed was that CtrA is required for RcGTA maturation and/or lysis. Manual lysis did not restore GTA-mediated transfer, expression of the endolysin gene was impaired when GafA was overexpressed in a ctrA knock-out background (qPCR data) and DNase insensitive DNA was not detected in the supernatant. These data indicate that GafA induces production of RcGTA particles and packaging of DNA but CtrA is also required for the RcGTA particles to become

infective and for the cells to lyse, which is consistent with previous work regarding deletion of ctrA. Text added to describe these new data.

- Line 84: The results presented here contradict the hypothesis presented in line 82. The fact that deletion of ctrA reduces expression of different genes indicates that ctrA is also involved (directly or indirectly) in the expression of the GTA structural genes.

There is no contradiction here. The regulatory model presented shows that CtrA is a global regulator required for GafA expression, among many other effects, and its deletion prevents GTA expression via GafA.

- Since crtA induces gafA transcription, it is not clear for this reviewer why overexpression of the crtA gene does not increase significantly transcription of the GTA structural genes. Is this suggesting additional levels of control (i. e. post-transcriptional or translational modifications)?

It is true that simple positive regulation by CtrA should mean that CtrA overexpression should lead to GafA and GTA overexpression and this is referred to in the text along with a comment about possible epigenetic control. There are a number of potential explanations e.g. precise expression levels, balance of phosphorylated vs non-phosphorylated CtrA, methylation etc. Additional text and data has been added indicating that the quorum sensing protein GtaR also binds to the GafA promoter close to the CtrA binding site, therefore, heterogeneity of quorum sensing response could also explain the phase variation observed for GTA expression.

- Some of the figures should include statistical analyses.

Amended

- To validate the experiments, the EMSA experiments should include competition with un-labelled DNA and competition with unrelated DNA.

Additional controls added

- More importantly, the ctrA binding sites should be mutated and the impact of these mutations in terms of GTA transfer should be evaluated.

The ctrA binding sites in the gafA promoter have now been mutated and changes to both half sites significantly decrease gene transfer frequency. New text and data added to the manuscript.

Reviewer #3 (Remarks to the Author):

This manuscript uses comparative transcriptomics to identify a likely key regulator of the life cycle of gene transfer agents (GTAs). GTAs represent a fourth mechanism of horizontal gene transfer agent in bacteria and are interesting from an evolutionary perspective in that they do not transfer their own DNA, but rather randomly package small segments of host DNA and transfer them to other members of the same species.

The manuscript identifies GafA as a direct activator of the GTA, and uses a number of techniques to investigate the relationship between GafA, CtrA and several other transcription factors known to be involved in GTA lifecycle. The manuscript concludes with a proposed model of GTA regulation, which attempts to bring together a number of observations from the literature including the roles of GtaR, CtrA, ClpXP, LexA and various phosphorelay components.

Statistics seem appropriate.

The results of this study appear to be novel and would clearly be of interest to others in the microbiology/evolution community.

Although the results are intriguing, I felt that stronger biochemical evidence for GafA binding, and

some pursuit of identification of its binding site is required, as this is a major claim of the manuscript. A relatively large segment of DNA (633bp) is used in the band shift assay (Fig 3D). It isn't made clear (i) how well the gafA promoter sequence is defined, (ii) where the promoter is located within this 633 bp fragment, or (iii) why this particular sized fragment was chosen.

The original fragment used material already available in the lab and was intentionally large because the binding site was unknown. Additional text has been provided to explain the rationale.

Although an MBP-gafA fusion protein is used out of necessity, further experiments could be performed relatively easily here. Why not use progressively smaller DNA fragments in the band shift assay, or ideally use DNase footprinting to more closely define the GafA binding site? The most likely location for GafA binding would seem to be close to the promoter, as is the case for CtrA. It would be particularly interesting to see the relative positions of the CtrA and GafA (and GtaR); see below) binding sites at the pGafA promoter.

Additional data to narrow down the binding site has now been added. Increased detail has also been added to the promoter regions including detailed transcript abundance from the RNAseq to predict the TSS

Two shifted bands are seen in the GafA band shift assay. With such a large fragment of DNA, can it be excluded that there are two independent sites on the fragment. Is there a biochemical explanation for the presence of two bands?

This figure has now been updated

Similarly, a useful experiment to provide stronger evidence for GafA as a key regulatory protein might be to make a mutation at one or two key residues within the HTH motif (mutating to a residue that is often found at this position) and see if binding is lost in the band shift assay.

This is an interesting suggestion. The data presented clearly show DNA binding and a phenotypic effect of GafA overexpression, but biochemical/structural data for the protein could certainly be the basis for a future paper

The CtrA band shift assays (Fig 3B) use a 50bp fragment and two shifted bands are annotated on the figure. Again is there biochemical explanation? There is a third band (lower mobility) which appears in the first three lanes with low concentrations of ctrA present. Is this significant? – whether this band is also present in the no protein control lane is a little hard to see due to bleed through on the left.

The figure has been updated. The lower band is not significant because it is present in the negative control, which is clearer in the new version. The CtrA binding site is made of two half sites and CtrA is known to bind to a short form of this sequence (TTAA) therefore multiple bands are likely to be partial or multiple occupancy.

Some discussion of the role of sigma factors in GTAs would be appropriate. Mercer et al (2014) (ref 23 in the manuscript) explored the effects of knocking out of each of the non-essential host sigma factors. No impact on GTA was observed. Given the rpoE domain found in GafA, could GafA act as an alternative sigma factor?

Text added to address this. It is possible that GafA is acting as a sigma factor and more work will be carried out in the future to address this more concretely. The binding site is close to the -10 promoter element and TSS. Additional discussion of Mercer et al has also been added.

The results will no doubt influence thinking in this field. A paper as recent as June 2018 (Appl Environ Microbiol 84:e00275-18. <https://doi.org/10.1128/AEM.00275-18>) states that the transcription factors acting on GTA promoters remain a mystery, indicating the work here is

important. I note that those authors also state that they saw no CtrA binding around GTA promoters and refer to unpublished band shift assays. This is in contrast to the strong binding reported in the present manuscript. Hence I feel there is a need to present the best evidence possible for specific DNA binding in the manuscript here.

Agreed. The evidence has been improved in the provided figures. I am aware of the work carried out in the paper referred to but it is difficult to say there is a contradiction because the promoters tested were not stipulated in the paper and the EMSAs were not presented. It's important to note that GafA was not considered to be a GTA-related gene until recently and the first author of the paper referred to left the lab a couple of years ago, so it is possible they did not test it or that the conditions used were not sensitive enough e.g. EMSAs in previous papers from the lab use large DNA fragments and ethidium staining for detection.

Figure legends could be made more informative. For example, to specify what size fragments are used for the gel shifts – partially described in methods but some use short oligos, other use much longer (633bp) DNA sequences, others eg endolysin promoter (fig S5B) not described. Panels often seem to be presented out of order in the legends

The figure legends have now been updated

Fig 3A – It would be helpful for the reader to show other promoter features here, if known (eg RNAP binding sequences).

This figure has now been updated

Fig. S1 – The meaning of the HTH alignment below the HTH motif wasn't clear.

This figure and legend have now been updated

Fig. S6. Predicted GtaR binding sites at the endolysin promoter and gafA promoter. Indicate where these are in the context of the promoter. What is the position of the proposed GtaR site at pGafA relative to the CtrA binding site shown in Fig S4? Given GtaR is a key protein in GTA production (Fig 4) the author might also consider providing direct in vitro evidence (band shift/footprint) for GtaR binding at the proposed site.

Agreed. This figure has now been deleted. GtaR was purified and additional data has been provided that it binds to the same 50 bp oligo in the GafA promoter as CtrA. Additional detail has also been added to the promoter diagram to incorporate transcription data, promoter features and binding sites.

REVIEWERS' COMMENTS:

Reviewer #1 (Remarks to the Author):

My comments on the previous version of the manuscript were mostly addressed. Two minor issues remain:

Comment #3. I still think that discussion of antibiotic resistance (e.g. statements like "Infections with AMR bacteria are an established threat to human health and, if left unchecked, deaths per year are projected to rise from ~700,000 in 2016 to 10 million by 2050 with associated economic costs running into trillions of dollars") at the very beginning of the manuscript is irrelevant to the main findings of the paper, and is somewhat distracting. Perhaps, a better place for a speculation regarding possible GTA involvement in antibiotic resistance transfer is at the end of the manuscript. But this is my subjective opinion, with which the author disagrees in his rebuttal. I will leave this issue for the editor to resolve.

Comment #33. I think locus tags (or perhaps accession numbers) should be provided for all genes listed in the Additional Table 1, even if they have gene names like "rpsR". This will help readers to identify genes in question faster. The response to reviews indicates that the table was amended, but the provided table does not include the information. May be my original request was not clearly stated.

Reviewer #2 (Remarks to the Author):

I'm very satisfied with this new version of the manuscript, and I must congratulate the author because he addressed and/or discussed brilliantly all the questions raised previously.

There is, however, a problem here, related with the comparison between transduction and GTA-mediated transfer. I do not agree that the number of genes mobilised by transduction are far less frequent than those mobilised by GTAs. While in generalised transduction the proportion of the transducing particles is small compared to the number of phage particles, in absolute terms that number is quite important. So the bias is not important here because generalised transduction packaged all the bacterial genome at high frequency. More importantly, the bacterial DNA is packaged in fragments of 40-50 kb, which can be easily incorporated in the recipient strain by double recombination. The incorporation of the DNA mobilised via GTAs into the recipient cells is by far more difficult, since the fragments are very small. In fact, because of this important limitation, some authors do not consider the transfer of DNA as the main role of the GTAs.

Even I consider generalised transduction as a more powerful mechanism involved in gene transfer than the GTAs, as previously discussed, the recent discovery of lateral transduction (doi: 10.1126/science.aat5867) definitely establishes transduction as the main force driving bacterial evolution. All the cons raised by the author disappear here. This nice discussion should be properly incorporated into the manuscript.

Reviewer #3 (Remarks to the Author):

The author has performed a number of the suggested experiments (EMSA in particular), rearranged figures for clarity, and significantly amended the text in response to reviewer comments. This has improved the readability of the manuscript and has provided important additional supporting evidence for the regulatory network proposed.

I am satisfied that the author has addressed my concerns.

Some minor comments/typos:

Line 22: GafA binding should be high nanomolar rather than high millimolar

Title of Figure 6 is incorrect. Should be something like "GafA binding to the RcGTA cluster"

REVIEWERS' COMMENTS:

Reviewer #1 (Remarks to the Author):

My comments on the previous version of the manuscript were mostly addressed. Two minor issues remain:

Comment #3. I still think that discussion of antibiotic resistance (e.g. statements like “Infections with AMR bacteria are an established threat to human health and, if left unchecked, deaths per year are projected to rise from ~700,000 in 2016 to 10 million by 2050 with associated economic costs running into trillions of dollars”) at the very beginning of the manuscript is irrelevant to the main findings of the paper, and is somewhat distracting. Perhaps, a better place for a speculation regarding possible GTA involvement in antibiotic resistance transfer is at the end of the manuscript. But this is my subjective opinion, with which the author disagrees in his rebuttal. I will leave this issue for the editor to resolve.

The statements RE antibiotic resistance have been deleted.

Comment #33. I think locus tags (or perhaps accession numbers) should be provided for all genes listed in the Additional Table 1, even if they have gene names like “rpsR”. This will help readers to identify genes in question faster. The response to reviews indicates that the table was amended, but the provided table does not include the information. May be my original request was not clearly stated.

Locus tags added for all entries in tables 1 & 2

Reviewer #2 (Remarks to the Author):

I'm very satisfied with this new version of the manuscript, and I must congratulate the author because he addressed and/or discussed brilliantly all the questions raised previously. There is, however, a problem here, related with the comparison between transduction and GTA-mediated transfer. I do not agree that the number of genes mobilised by transduction are far less frequent than those mobilised by GTAs. While in generalised transduction the proportion of the transducing particles is small compared to the number of phage particles, in absolute terms that number is quite important. So the bias is not important here because generalised transduction packaged all the bacterial genome at high frequency. More importantly, the bacterial DNA is packaged in fragments of 40-50 kb, which can be easily incorporated in the recipient strain by double recombination. The incorporation of the DNA mobilised via GTAs into the recipient cells is by far more difficult, since the fragments are very small. In fact, because of this important limitation, some authors do not consider the transfer of DNA as the main role of the GTAs.

Even I consider generalised transduction as a more powerful mechanism involved in gene transfer than the GTAs, as previously discussed, the recent discovery of lateral transduction (doi: 10.1126/science.aat5867) definitely establishes transduction as the main force driving bacterial evolution. All the cons raised by the author disappear here. This nice discussion should be properly incorporated into the manuscript.

Additional text in the introduction added to address the reviewer's concerns.

Reviewer #3 (Remarks to the Author):

Revised version NCOMMS-18-25120A The author has performed a number of the suggested experiments (EMSAs in particular), rearranged figures for clarity, and significantly amended the text in response to reviewer comments. This has improved the readability of the manuscript and has provided important additional supporting evidence for the regulatory network proposed. I am satisfied that the author has addressed my concerns. Some minor comments/typos: Line 22: GafA binding should be high nanomolar rather than high millimolar.

Corrected

Title of Figure 6 is incorrect. Should be something like "GafA binding to the RCGTA cluster".

Corrected